# A nanoengineered topical transmucosal cisplatin delivery system induces anti-tumor response in animal models and patients with oral cancer

Manijeh Goldberg [1,2,3,4,17 ✉], Aaron Manzi[1,3,4,17], Amritpreet Birdi[4], Brandon Laporte[4], Peter Conway[4], Stefanie Cantin[4], Vasudha Mishra[5], Alka Singh[5], Alexander T. Pearson [5], Eric R. Goldberg[4], Sam Goldberger[4], Benjamin Flaum[4], Rifat Hasina[6], Nyall R. London[7,8], Gary L. Gallia[7,8], Chetan Bettegowda [9], Simon Young[10], Vlad Sandulache[11], James Melville[10], Jonathan Shum[10], Sonya E. O'Neill[2,12], Erkin Aydin [1], Alex Zhavoronkov [13], Anxo Vidal [14], Atenea Soto[14], Maria Jose Alonso[14], Ari J. Rosenberg[5], Mark W. Lingen[15], Anil D'Cruz[16], Nishant Agrawal [6 ✉] & Evgeny Izumchenko [5 ✉]

Despite therapeutic advancements, oral cavity squamous cell carcinoma (OCSCC) remains a difficult disease to treat. Systemic platinum-based chemotherapy often leads to dose-limiting toxicity (DLT), affecting quality of life. PRV111 is a nanotechnology-based system for local delivery of cisplatin loaded chitosan particles, that penetrate tumor tissue and lymphatic channels while avoiding systemic circulation and toxicity. Here we evaluate PRV111 using animal models of oral cancer, followed by a clinical trial in patients with OCSCC. In vivo, PRV111 results in elevated cisplatin retention in tumors and negligible systemic levels, compared to the intravenous, intraperitoneal or intratumoral delivery. Furthermore, PRV111 produces robust anti-tumor responses in subcutaneous and orthotopic cancer models and results in complete regression of carcinogen-induced premalignant lesions. In a phase 1/2, open-label, single-arm trial (NCT03502148), primary endpoints of efficacy (≥30% tumor volume reduction) and safety (incidence of DLTs) of neoadjuvant PRV111 were reached, with 69% tumor reduction in ~7 days and over 87% response rate. Secondary endpoints (cisplatin biodistribution, loco-regional control, and technical success) were achieved. No DLTs or drug-related serious adverse events were reported. No locoregional recurrences were evident in 6 months. Integration of PRV111 with current standard of care may improve health outcomes and survival of patients with OCSCC.

---

A full list of author affiliations appears at the end of the paper.

Oral cavity squamous cell carcinoma (OCSCC), the most common subtype of head and neck squamous cell carcinoma (HNSCC), is a devastating disease, causing substantial morbidity and mortality. While clinical outcomes for patients with OCSCC have improved over the last two decades, the prognosis remains relatively unfavorable, with 5-year overall survival (OS) rate hovering at ~70-80% even for early stage (T1-T2) tumors[1].

The conventional management for newly diagnosed, locally advanced OCSCC includes surgical resection with or without adjuvant radiotherapy or chemoradiotherapy[2–5], depending on an assessment of high-risk features. Despite advancements in the conventional therapeutic approaches, surgical resection can lead to permanent functional deficits, and negatively affect a patient's ability to speak, chew, and swallow[6]. Social isolation is common following treatment and, as a result, patients with oral cancers have one of the highest depression and suicide rates[6,7]. Surgery may be followed with adjuvant chemoradiotherapy, with cisplatin (cis-diamminedichloroplatinum(II), CDDP) being the most frequently administered systemic chemotherapeutic agent[8,9]. Intracellularly, cisplatin acts primarily by producing DNA inter- and intra-strand crosslinks, which prevent DNA replication and promote apoptosis of tumor cells[10]. However, given the nonselective targeting of both healthy and malignant tissues, its clinical utility is hindered by adverse effects and toxicities, including protracted nausea, vomiting, ototoxicity, acute nephrotoxicity, myelosuppression, and chronic neurotoxicity[11,12]. These side effects of systemic cisplatin administration can be dose-limiting, reducing the cumulative dosage and length of treatment for patients, and subsequently limiting therapeutic benefits[13]. Thus, development of novel therapeutic delivery approaches which enhance efficacy and reduce toxicity is paramount to improve health outcomes and survival of patients with OCSCC.

The local treatment of OCSCC may significantly reduce or eliminate the systemic toxicities and adverse side effects traditionally associated with intravenous (IV) cisplatin chemotherapy. To this end, a wide range of nanoparticles (NPs) based drug delivery systems (DDSs) have been explored as alternative cisplatin delivery methods that may promote its accumulation and retention in cancer cells and thus overcoming the low therapeutic ratio of the free drug[14–16]. While some of these NP formulations have demonstrated promising preclinical results, and a few have entered clinical trials, none have been approved for treatment of human cancers and only a few studies have tested the potential utilization of these approaches in head and neck cancer models[17–21].

The difficulty of maintaining the delicate balance of extracellular stability and intracellular drug uptake remains a major barrier that hinders the application of polymeric cisplatin-loaded NPs in oncology. To overcome this challenge, we have encapsulated cisplatin within chitosan nanoparticles (a non-toxic, biocompatible, biodegradable polysaccharide derived from natural chitin[22]) and embedded them within a chitosan sponge matrix (CSM)[17]. Such encapsulation protects cisplatin from biological deactivation and promotes increased rates of cell uptake[23]. Additionally, using cationic chitosan as the polymer for both the NPs and CSM, allows for electrostatic interactions with anionic domains of mucin proteins in the oral cavity[23]. This mucoadhesive property retains NPs, and subsequently cisplatin, at the delivery site[17], facilitating topical local application.

Building on this previous work by our group and others[17,22], we have developed PRV111, a self-adhesive cisplatin transmucosal system designed to deliver cisplatin-loaded chitosan particles (CLPs) to anatomically accessible oral cancers, such as lip, tongue, gum, floor of mouth, gingiva, buccal mucosa, and other locations in the oral cavity. PRV111 is a thin, 2-layer, matrix-type, polymeric transmucosal patch, consisting of a chitosan matrix layer embedded with CLPs and an impermeable ethyl-cellulose adhesive backing, designed to provide targeted drug delivery and prevent CLP washout from saliva. Each PRV111 topical patch contains 0.5 mg/cm$^2$ (2 mg) of cisplatin, and covers a tumor region of 4 cm$^2$. The system also incorporates a permeation enhancer (PE), which functions by reversibly opening the tight junctions between the cells, allowing for optimal penetration and absorption of the CLPs released from the patch. The released CLPs swell to approximately 0.5 micron when exposed to moisture, allowing them to diffuse across the porous matrix and into the tumor tissue. These particles are too large to penetrate into the vasculature (vasculature pore size is 2–15 nm), and therefore there is no systemic cisplatin exposure[24].

In the current work we first examine safety and efficacy of the PRV111 using several well-established animal models for studying oral mucosal carcinogenesis, demonstrating the superior local retention, efficacy, and safety of the CLPs compared to either non-encapsulated or intravenous cisplatin administration in all models tested. Specifically, PRV111 is tested in athymic nude mice bearing a human oral cancer cell line (FaDu) induced subcutaneous tumor[25], in an orthotopic HCPC-1 cell line induced cheek pouch tumor in Golden Syrian hamster[26], and in a cheek pouch precancerous lesions induced by the 7,12-Dimethyl-benz(a)-anthracene (DMBA) exposure[27–33]. Following the proof-of-concept in vivo studies, a pilot phase I/II in-human clinical trial is designed to improve efficacy and reduce toxicity in patients with resectable OCSCC by neoadjuvant local administration of PRV111. Taken together, this study indicates that PRV111 nanoengineered cisplatin patch may result in larger and more effective doses of cisplatin delivered to the tumor, significantly reducing, and potentially eliminating altogether, the systemic toxicities associated with intravenous cisplatin, while inducing a robust anti-tumor response. Given its promising clinical performance and attractive safety profile, local treatment with PRV111 may provide a safer and more effective alternative to intravenous chemotherapy, improving outcomes and quality of life while lowering the socio-economic burden of OCSCC.

## Results

**Cisplatin chitosan nanoparticles (CDDP-NP) demonstrate improved local drug retention and induce potent anti-tumor response in FaDu tumor bearing mice xenografts.** To investigate whether chitosan encapsulated cisplatin NPs (CDDP-NPs) could increase the efficacy of cisplatin and decrease the associated toxicities in vivo, athymic nude mice were inoculated subcutaneously with human HNSCC FaDu cells, randomized into 5 groups ($n = 6$) two weeks after inoculation, and treated with intratumoral (IT) injection of vehicle control (PBS-IT), IT injection of free cisplatin (CDDP-IT), intravenous (IV) free cisplatin (CDDP-IV), IT application of the CDDP-NP, and IT administration of blank NPs without cisplatin encapsulation (BLK-NP) (Fig. 1A, Supplementary Fig. 1A). During the treatment course, tumor volumes as well as body weight were assessed twice a week as described in the Methods section. Animals in the CDDP-IV group were euthanized at day 7 after treatment initiation due to the rapidly decreasing body weight and signs of toxicity associated with high level of systemic CDDP circulation (Fig. 1B, E). Notably, while comparable effects on tumor growth inhibition were observed in CDDP-IT and CDDP-NP groups during the first 12 days of treatment, CDDP-NP demonstrated a further decrease in the tumor volumes, resulting in near complete tumor regression by day 18 (Fig. 1B, Supplementary Figure 2). In contrast to CDDP-NP, tumor inhibitory effect of CDDP-IT was abrogated, and by the endpoint, tumors volumes in this group were not substantially different from tumor treated with either PBS-IT or BLK-NP (Fig. 1B). As the potent antitumor effect of

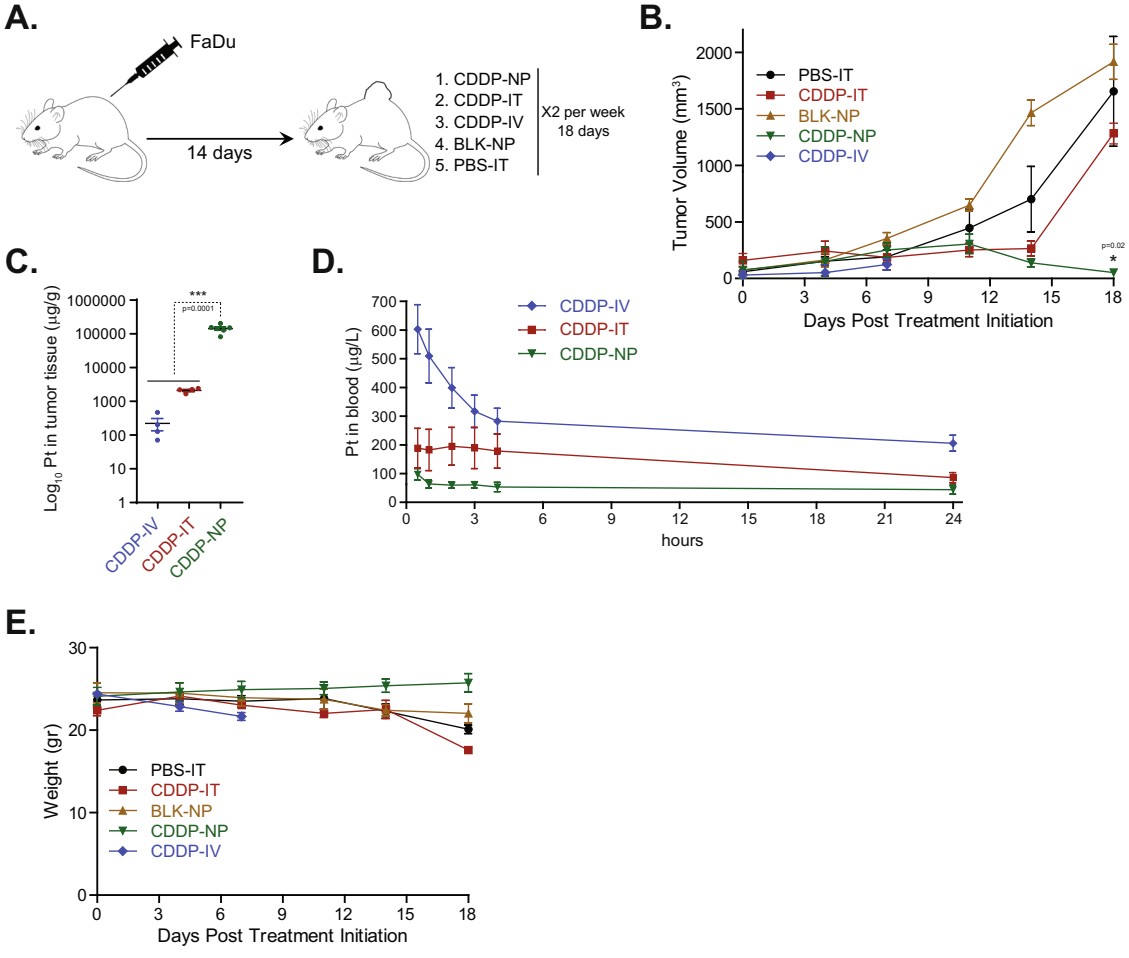

**Fig. 1 Cisplatin chitosan nanoparticles (CDDP-NP) demonstrate improved local drug retention and induce potent anti-tumor response in FaDu tumor bearing mice xenografts. A** Schematic representation of the experiment. **B** Nude mice xenografts bearing subcutaneous human FaDu HNSCC cell line induced tumors were treated with either PBS-IT, CDDP-IT, CDDP-IV, BLK-NP, or CDDP-NP. Graphs show the average tumor volume for 6 animals per group ± SEM. P-value was performed by a 1-sided Wilcoxon Rank-sum Test. Asterisk represents statistical significance between the CDDP-IT and CDDP-NP groups ($p < 0.05$). **C** Residual tumors were harvested after 4 treatments (day 14) for CDDP-NP ($n = 6$) or CDDP-IT ($n = 4$) and 2 treatments (day 7) for CDDP-IV ($n = 4$). Samples were weighted, homogenized and cisplatin level was quantified by ICP-MS, shown as average ± SEM ($p < 0.0001$; Student's unpaired $t$-test). **D** Blood was collected from 4 animals per group after administration of the first dose at indicated time points and level of cisplatin was quantified by ICP-MS, shown as average ± SEM. **E** Graph shows the average body weights for 6 animals per group ± SEM. Source data are provided as a Source Data file.

CDDP-NP may be due to its ability to remain in the tumor without rapid permeation to the vasculature, we have quantified platinum (Pt) concentration in both tumor and blood samples by inductively coupled plasma mass spectrometry (ICP-MS)[34–37]. Analysis of residual tumors collected following 4 treatments with either CDDP-NP or CDDP-IT and 2 treatments with CDDP-IV ($n \geq 4$ animals per group), confirmed a substantially higher cisplatin level in tumors that received an IT injection of CDDP-NP (Fig. 1C). On the other hand, a pharmacokinetic (PK) analysis performed after administration of the first dose, revealed that mice in the CDDP-NP group showed a significantly lower blood level of cisplatin, compared to either the CDDP-IT or CDDP-IV treated animals (Fig. 1D). Furthermore, our results show that administration of chitosan CDDP-NPs was well tolerated in the animals, with no overt toxicity (as measured by morbidity or significant body weight loss) (Fig. 1E).

**Local application of PRV111 induced robust anti-tumor response in hamster orthotopic oral cancer model.** Given the encouraging results demonstrated by intratumoral CDDP-NP in immunocompromised mice, we next assessed the effect of

PRV111 patch drug delivery system using an orthotopic immunocompetent hamster oral cancer model (Supplementary Fig. 1B). Oral cancers were induced by inoculating HCPC-1 epidermoid carcinoma cell line into the cheek pouch of golden Syrian hamsters. Twelve days after inoculation, animals were randomized into 4 groups ($n = 6$) and treated with PRV111 patch, intraperitoneal injection of free cisplatin (CDDP-IP), drug-free patch (BLK-patch), and IP injection of vehicle control (PBS-IP) (Fig. 2A, Supplementary Fig. 1B). Hamsters were treated for 17 days (6 treatments total). Both PRV111 and BLK patches were administered to the tumor 5 min after PE application and kept for 1 h under anesthesia (Supplementary Fig. 3A). Oral cavity tumor volumes as well as weight were examined for 28 days post treatment initiation. Animals treated with PRV111 demonstrated substantial reduction in tumor sizes after just 3 treatments (day 10) (Fig. 2B), with complete tumor regression achieved in most animals by day 17 (after 5 treatments with PRV111) (Supplementary Fig. 3B). Notably, all hamsters in the PRV111 group remained tumor progression free 10 days post treatment termination (Fig. 2B). Compared to PRV111, CDDP-IP remained ineffective during the treatment period, with tumor sizes having

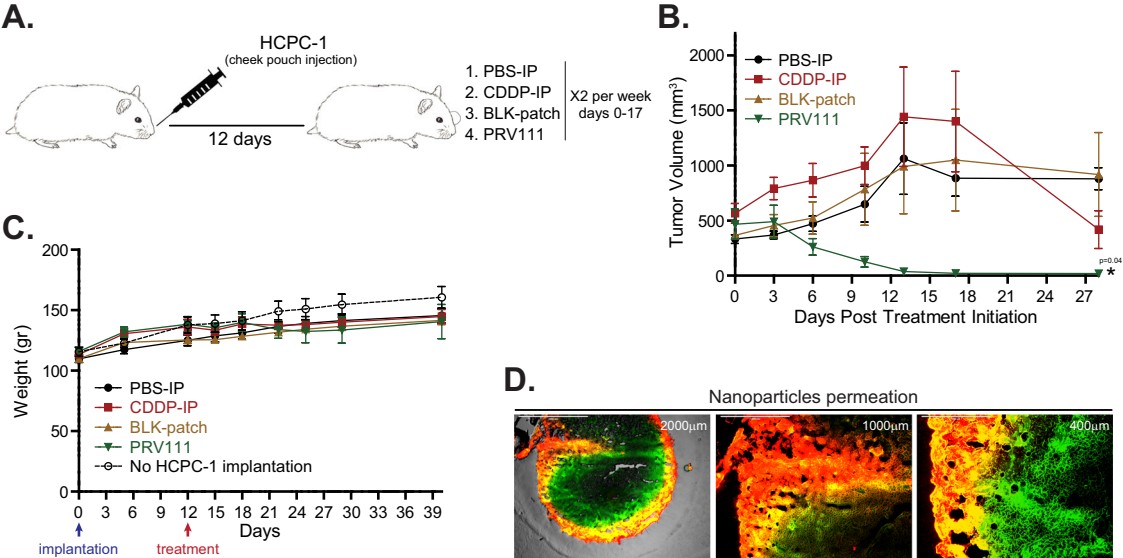

**Fig. 2 Local application of PRV111 induced robust anti-tumor response in hamster orthotopic oral cancer model. A** Schematic representation of the experiment. **B** Golden Syrian hamsters bearing orthotopic tumors induced by injection of HCPC-1 cell line into the cheek pouch were treated with either PBS-IP, CDDP-IP, BLK-patch or PRV111. Graphs show the average tumor volume for 6 animals per group ± SEM. *P*-value was performed by a 1-sided Wilcoxon Rank-sum method. Asterisk represents statistical significance between the CDDP-IP and PRV111 groups (*p* < 0.05). **C** Graph shows the average body weights for 6 animals per group ± SEM. Body weights of 5 tumor free animals without treatment were measure along the tumor bearing counterparts (dashed line). **D** Representative fluorescence images of tumor sections at indicated magnification taken after treatment of the hamster with PRV111 patch containing chitosan particles labeled with Cy5 (red) and encapsulating FITC (green). Yellow areas display dual-labeling, NPs with encapsulated FITC. Permeation experiment was repeated in 6 tumors. Source data are provided as a Source Data file.

no notable differences from PBS-IP and BLK-patch treated groups. However, significant tumor shrinkage occurred during the last week prior to the experimental endpoint, suggesting inadequate CDDP accumulation in the tumor tissue. No significant difference in body weight was observed between tumor bearing animals in four treatment groups and normal (tumor free) littermates (Fig. 2C). To evaluate the penetration depth of topically delivered drug, tumor bearing animals were treated with PRV111 patch loaded with Cy5 conjugated chitosan nanoparticles (red color) encapsulating FITC (green color). After the patch was removed, tumors were harvested, washed to clean the excess material that did not permeate, sectioned and analyzed using fluorescent microscopy. Dye-labeled nanoparticles successfully permeated through the lesion's surface, with the carried drug (FITC) widely distributed throughout the entire tumor mass (Fig. 2D). Note, nanoparticles loaded to PRV111 were designed to penetrate and absorb only in areas where PE (a critical component of the PRV111 system) is applied, thus minimizing the damage to the adjacent non-tumorous tissue (Supplementary Fig. 4). Due to its close resemblance to human buccal mucosa[38,39], porcine oral mucosa was used for this experiment.

**Topical administration of PRV111 reduced cisplatin associated toxicities.** Analysis performed one hour after administration of the first dose demonstrated that blood level of cisplatin in the CDDP-IP group was over 20-fold higher than the level detected in PRV111 treated animals (Fig. 3A). At the end-point, hamsters were sacrificed and major organs such as liver, lungs, kidneys, spleen, brain and heart, as well as the tongue, healthy contralateral cheek pouch, and tumors, were isolated to evaluate the tissue distribution of cisplatin in PRV111 and CDDP-IP treated groups using ICP-MS. Note, we were able to collect very small residual tumors samples from four animals treated with PRV111, as the remaining hamsters were tumor free upon necropsy. A significantly higher cisplatin level was observed in the tumor tissue collected from PRV111 treated animals, compared to the

tumors treated with CDDP-IP delivery (Fig. 3B). Other tissues harvested from the PRV111 group showed negligible cisplatin levels, whereas treatment with CDDP-IP resulted in a higher drug uptake by all organs tested (Fig. 3C). Analysis of histological sections revealed a substantial kidney tubular necrosis in animals treated with CDDP-IP (Fig. 3D), while the PRV111 group showed no damage to any major organs, including the kidney.

**Treatment with PRV111 prevents tumor recurrence in vivo.** To assess durability of response, six animals that achieved complete tumor regression after receiving a full course of PRV111 therapy or IP cisplatin injections were selected and monitored long-term for recurrence and survival. Three hamsters from each group were planned to be harvested 9 months post treatment cessation for toxicology assessment and necropsy. However, 5 of the 6 CDDP-IP treated animals (83%) recurred prior to the 9-month mark: two recurrences were detected at 4 months, one at 5 months, and two at 8 months (Fig. 3E). All PRV111 treated animals were tumor free, and no remarkable findings were noted upon histological examination of major organs following necropsy. The remaining animals (3 PRV111 treated hamsters and 1 animal that received CDDP-IP) remained tumor free until the end of life (Fig. 3E).

**PRV111 induced rapid and sustainable regression of carcinogen-induced oral premalignant lesions.** Many OCSCCs arise from an existing premalignant oral dysplastic lesion, often visually identified as leukoplakia or erythroplakia[40]. While currently surgical excision and cryotherapy are the most frequent treatment options, as oral premalignant lesions are readily accessible, several topical treatment alternatives for oral cancer chemoprevention have been tested in the past decades[41,42]. However, all of these methods were associated with recurrence of the disease[42]. As PRV111 showed encouraging safety profile in vivo, we next evaluated its ability to promote regression of the

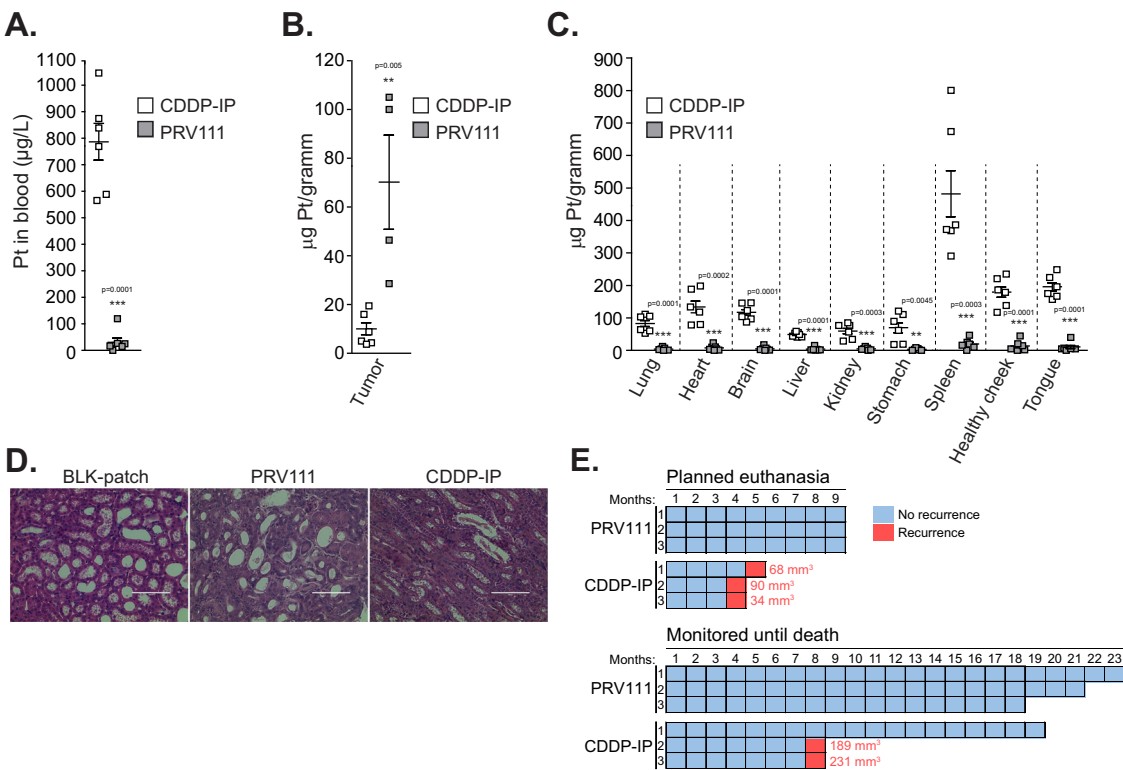

**Fig. 3 Topical administration of PRV111 reduced cisplatin associated toxicities and prevented tumor recurrence in vivo. A** Blood was collected after administration of the first dose of CDDP-IP or PRV111 (6 animals per group), and level of cisplatin was quantified by ICP-MS, shown as average ± SEM ($p = 0.0001$; Student's unpaired $t$-test). **B, C** Hamsters treated with CDDP-IP or PRV111 (6 animals per group) were sacrificed and residual tumors as well as lungs, heart, brain, liver, kidneys, stomach, spleen, healthy contralateral cheek pouch, and tongue were harvested. Samples were weighted, processed, and biodistribution of cisplatin was assessed by ICP-MS, shown as average ± SEM ($p = 0.005$ for tumor tissue all body organs analyzed; Student's unpaired $t$-test). Note, residual tumors were collected from four animals treated with PRV111. **D** Representative H&E stained histopathological images of kidneys (×200; scale bar—100 µm). Three tumors per group were stained. **E** Animals that were tumor free after treatment with either PRV111 or CDDP-IP (6 per group) were monitored for recurrence and survival during the indicated time period (months) post treatment cessation. Animals were evaluated and weighted weekly until day 45, and once a month thereafter. Cyan squares indicate a month when animal remained tumor-free. Pink squares indicate a month when recurrence was detected. Tumor volume upon recurrence is provided next to the each pink square. Source data are provided as a Source Data file.

premalignant lesions using a hamster carcinogen-induced model (Supplementary Fig. 1C). Cheek pouches were painted with DMBA for 4 weeks to induce oral dysplasia (Supplementary Fig. 5A), randomized into 3 groups ($n = 3$) and treated with either PRV111, CDDP-IP, or drug free BLK-patch for two weeks (Fig. 4A). Presence of dysplastic lesions and body weight was periodically assessed for 49 days. While BLK-patch treated animals displayed unifocal oral dysplastic lesions throughout the entire monitoring period, all hamsters in PRV111 group were lesion-free after 3 treatments (day 14) (Fig. 4B). At the same time point, all CDDP-IP treated animals were positive for leukoplakia, with one animal showing detectable dysplasia even after 4 treatments (day 18). Furthermore, all PRV111 treated animals remained disease-free 35 days post treatment termination, whereas one of the three CDDP-IP treated hamsters was still harboring a precancerous neoplasm (Fig. 4B). There was no inflammation or other damage detected at the site of the PRV111 application (Supplementary Fig. 5B), and no local toxicity was observed in any other area of the oral cavity, including the tongue and or other cheek pouch. While animals showed no signs of toxicity and the body weights were not significantly different between the groups (Fig. 4C), a higher amount of cisplatin was detected in the blood one hour after the last CDDP-IP administration, compared to the levels detected in the PRV111 treated animals (Fig. 4D). Furthermore, a biodistribution analysis

revealed high levels of cisplatin in organs harvested from the CDDP-IP treated animals, whereas cisplatin levels in organs of the PRV111 treated hamsters were several fold lower (Fig. 4E). Analysis of kidney sections revealed extensive tubular necrosis in animals with intraperitoneal cisplatin treatment (Fig. 4F), compared to either PRV111 treated or control subjects.

**A phase I/II safety and efficacy study of PRV111 as a neoadjuvant therapy for early-stage OCSCC.** Based on a preclinical data indicating that PRV111 is an effective and safe method for delivering local cisplatin with no systemic toxicity, an open-label, single arm, 2-stage adaptive[43–45] phase I/II clinical trial (NCT03502148) was designed to further evaluate its safety and efficacy for treating OCSCC (see Methods for study design). Ten patients with confirmed (T1-T2, Nx, M0 [AJCC 7th Edition]) OCSCC and tumor size ≤ 4.0 cm were enrolled. Demographic characteristics as well as alcohol and tobacco consumption history of the patients participated in this study are summarized in Tables 1 and 2, respectively. Three weeks prior to surgery, patients were administered one cycle of neoadjuvant PRV111, consisting of four treatment visits (Fig. 4A) (each visit dose: 6 mg or 12 mg of cisplatin depending on the tumor size and shape). During a monitoring visit, it was determined that one subject's tumor stage and size was outside of the eligibility criteria.

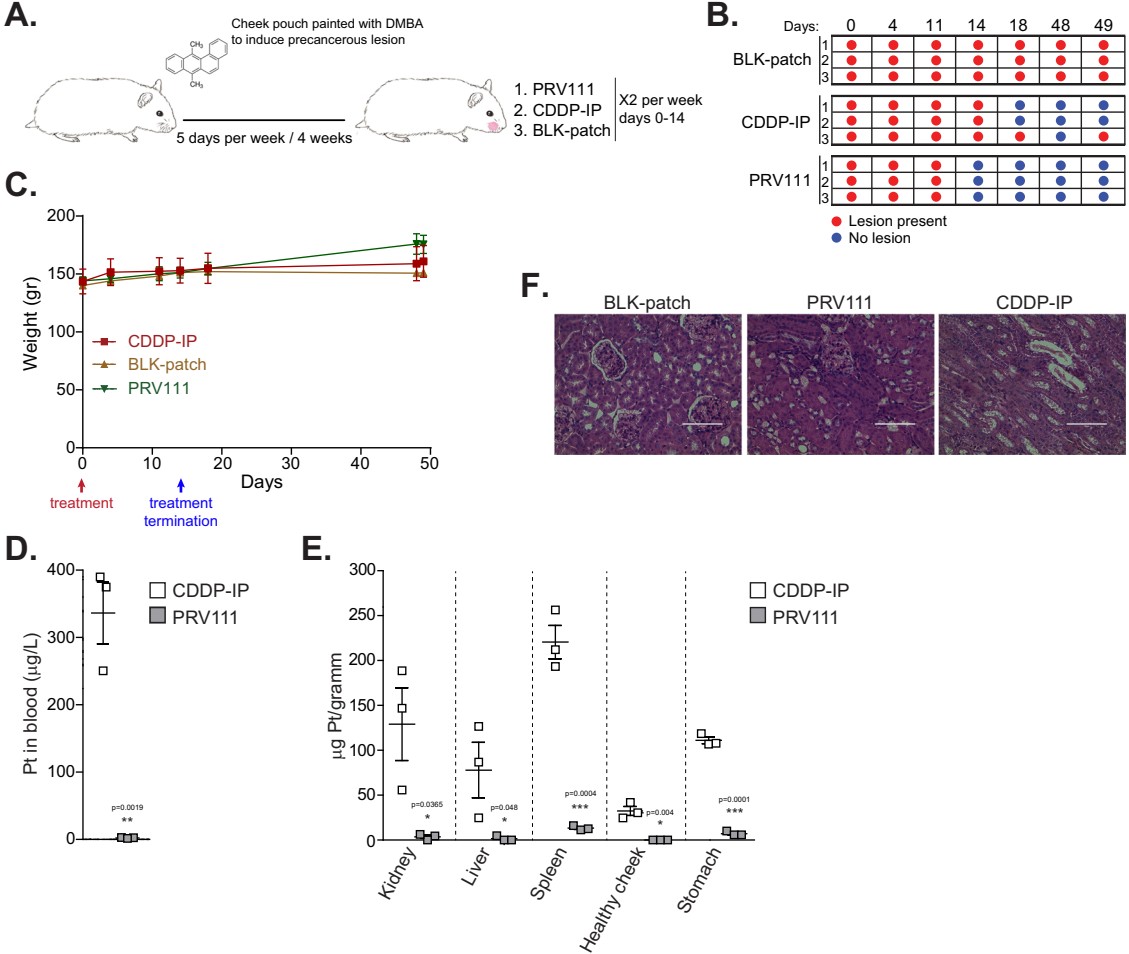

**Fig. 4 PRV111 induced rapid and sustainable regression of carcinogen-induced oral premalignant lesions. A** Schematic representation of the experiment. **B** Golden Syrian hamsters bearing DMBA-induced oral dysplastic lesion were treated with either BLK-patch, CDDP-IP or PRV111 (3 animals per group). Red circle indicates presence of the premalignant lesion, whereas blue circle indicates that the premalignant lesions was undetected. **C** Graph shows the average body weights for the 3 hamsters in each indicated group ± SEM. **D** Blood was collected from 3 animals per group after administration of the first dose of CDDP-IP or PRV111, and level of cisplatin was quantified by ICP-MS, shown as average ± SEM ($p < 0.0019$; Student's unpaired $t$-test). **E** Hamsters treated with CDDP-IP or PRV111 ($n = 3$) were sacrificed and indicated organs were harvested, weighted, processed, and biodistribution of cisplatin was assessed by ICP-MS, shown as average ± SEM ($p < 0.05$ for all body organs analyzed; Student's unpaired $t$-test). **F** Representative H&E stained histopathological images of kidneys (200×; scale bar—100 μm). Three tumors per group were stained. Source data are provided as a Source Data file.

| Table 1 Demographic characteristics of the enrolled patients. | |
|---|---|
| **Category** | **Patients (N = 10)** |
| Age (year) | |
| n | 10 |
| Mean (SD) | 64.3 (12.15) |
| Median | 62.5 |
| Min, Max | 49, 89 |
| Sex, n (%) | |
| Male | 5 (50.0) |
| Female | 5 (50.0) |
| Race, n (%) | |
| White | 8 (80.0) |
| Asian | 2 (20.0) |
| Ethnicity, n (%) | |
| Not Hispanic or Latino | 10 (100) |
| SD standard deviation. | |

Another subject had evidence of concurrent malignancy of the oral cavity and metastatic disease identified on the baseline CT examination. As such, these patients were excluded from the evaluable population, but included in safety analyses (safety population). Eight of the remaining subjects (two patients with T1 and 6 patients with T2 stage tumors) met the inclusion criteria, had evaluable biopsy samples, completed 4 treatments with PRV111, and had evaluable samples for the determination of cisplatin concentrations in tumor tissue, regional lymph nodes, and blood. All patients were human papillomavirus (HPV) negative by PCR targeting HPV16 or HPV18 subtype.

**PRV111 induced a rapid anti-tumor response in patients with locally advanced OCSCC.** A total of 182 patches were applied during the 4 treatment visits (Fig. 5A) in the 10 subjects in the safety population. None of the patches had peel issues and only 2.75% of patches were noted to have issues with adhesion during treatment (Supplementary Table 2). Average residual platinum remaining in the PRV111 patch following treatment application was consistent between subjects (Supplementary Table 3) and across applications (Supplementary Table 4) (ranging from 5.4%

| Table 2 Alcohol and tobacco consumption history of the enrolled patients. | |
| --- | --- |
| **Alcohol Consumption** | ***n* (%)** |
| Ever | 6 (60.0) |
| Never | 4 (40.0) |
| Number of years | |
| 16–20 years | 1 (10.0) |
| > 20 years | 4 (40.0) |
| Unknown | 1 (10.0) |
| **Tobacco products** | ***n* (%)** |
| Ever | 4 (40.0) |
| No tobacco use history | 6 (60.0) |
| Number of years | |
| 6–10 years | 1 (10.0) |
| > 20 years | 3 (30.0) |

to 13.4%), indicating sustained and uniform drug release. PRV111 successfully met all clinical primary endpoints, as well as safety and efficacy objectives. It caused 69% tumor volume reduction in ~7 days with over 87% response rate across 8 subjects in evaluable population (Fig. 5B, Supplementary Table 5). Notably, in five patients, tumor volume percent change from baseline was over 90% (Fig. 5B, C). In a single patient that progressed based on a plane measurement, PRV111 treatment caused the tumor to become more endophytic, making it difficult to measure a height of the tumor growing below the surface epithelium. The average time to tumor response was 5.5 days. The dynamics of total tumor volume change by visit is summarized in Supplementary Table 6. Post-surgery pathology reports indicated treatment effect in the form of necrosis/ulceration, cellular debris, and/or acute/chronic inflammation in some subjects (Fig. 5D, Supplementary Fig. 6) and negative margins in all cases. The PRV111 patch was well tolerated by all subjects, with no withdrawals from study due to adverse events (AEs). The most frequently reported AEs during treatment were Grade 1/Grade 2 oral pain (40%) and glossodynia (30%) (Supplementary Table 7). No dose-limiting toxicities, serious adverse events, or systemic toxicities (blood and urine tests were performed after each PRV111 administration) were reported (Supplementary Table 8), and no locoregional recurrences were evident in 6 months. Complete blood counts showed no evidence of myelosuppression (Supplementary Table 9), which is typically seen with IV cisplatin administration[46]. Expected local toxicity was observed on and around the tumor region due to acute and chronic necrosis and ulceration/blistering. However, all subjects were able to eat and drink after each treatment visit and no issues with healing were noted during follow-up. Pain was managed using Magic mouthwash or over-the-counter medication.

**PRV111 retained a high concentration of cytotoxic drug in the tumor and regional lymph nodes**. Total platinum levels were assessed in blood at each treatment visit, as well as in tumor and regional lymph node specimens collected post-treatment at surgery. Lymph node location, imaging details and metastatic status are summarized in Supplementary Table 10. The highest concentrations of cisplatin following application of PRV111 were found locally in tumor tissues (mean concentration 336.8 µg/g) and to a lesser extent in the first echelon lymph node basin (mean concentration 110 µg/g) (Supplementary Table 11). Systemic exposure to cisplatin was minimal, with maximum concentration (Cmax) in blood of 0.24 µM on average, ranging $0.06 - 0.62$ µM (Supplementary Table 11). Compared with concentrations of cisplatin that are typically achieved with IV standard of care

therapy[47,48], the concentration of cisplatin in tumor tissue and blood after treatment with PRV111 was 259x higher and 182x lower respectively (Fig. 5E). Interestingly, level of cisplatin in lymph nodes ablated from patients treated with PRV111 was over 80x higher than that achieved in tumors treated with IV cisplatin administration.

**PRV111 increased the number of tumor-infiltrating lymphocytes**. High levels of tumor-infiltrating lymphocytes (TILs) generally correspond with improved clinical outcome and overall survival in patients with HNSCC[49–51]. Preclinical studies and clinical evidence suggest that aside from its direct cytotoxic effect, cisplatin may induce antitumor immunity by enhancing numbers and function of TILs[52,53]. However, antitumor immunomodulatory effects of systemic cisplatin on tumor microenvironment (TME) are limited by bone marrow suppression, leading to abnormal hematopoiesis. Given the negligible amounts of cisplatin detected in systemic circulation following PRV111 administration (Fig. 5E, Supplementary Table 11), a concentrated CDDP delivery by PRV111 may favorably modulate the immune system in the TME while avoiding cisplatin-induced bone marrow dysfunction. To this end, levels of TILs were evaluated in pretreatment tumor biopsies and post-treatment surgical specimens. Notably, immunohistochemical analysis revealed significantly higher amounts of CD3 +, CD4 +, and CD8 + T-cells in post treatment slides compared with the pretreatment biopsies (Fig. 6), suggesting that local exposure to highly concentrated potent cisplatin may enhance the recruitment of immune effectors into the TME, contributing to patients' response in combination with its direct cytotoxic effects. Future studies aimed at delineating the balance of TILs with anti-tumor and immunosuppressive activity are necessary to further assess the clinical relevance of these observations.

**Discussion**
The multimodality treatment approach for locally advanced OCSCC has evolved over the last few decades, contributing to improvements in functional and esthetic outcomes[54]. However, despite these important advances, therapeutic options remain limited, and complications associated with surgical cancer ablation, long-term effects of radiation, and dose-limiting side effects of systemic cisplatin administration, still have a significant impact on the quality of life in patients with OCSCC[54].

Several strategies for local cisplatin delivery, including polymeric NP drug carriers, have been developed to improve the OCSCC treatment efficacy and ameliorate systemic toxicities induced by intravenous administration[14,16,21,55–60]. While some nanocarrier-based cisplatin delivery systems demonstrated better tolerability and improved therapeutic efficacy in vivo[21,61–63], only a few were tested in clinical trial settings[64,65] and none have achieved FDA-approval. Among the challenges that hinder clinical applicability of polymeric mucosal drug delivery systems (DDSs), is their limited ability to penetrate the epithelium deep enough to eliminate cancer cells remaining underneath superficial tumor layers[66,67]. This is in part due to a rapid washout by the saliva, clearance via the vasculature, and difficulty maintaining a concentrated drug release.

Chitosan, a natural polysaccharide obtained by alkaline deacetylation of chitin, is biodegradable, biocompatible, mucoadhesive, and a non-toxic polymer[68]. These excellent biological properties coupled with enhanced permeability, slow degradation, high hydrophobic drugs entrapment efficiency, and NPs stability[69–73], make chitosan a good candidate for developing non-invasive local DDSs, suitable for mucosal cancers, such as OCSCC[22,68,74–76]. While the in vivo anti-cancer efficacy of

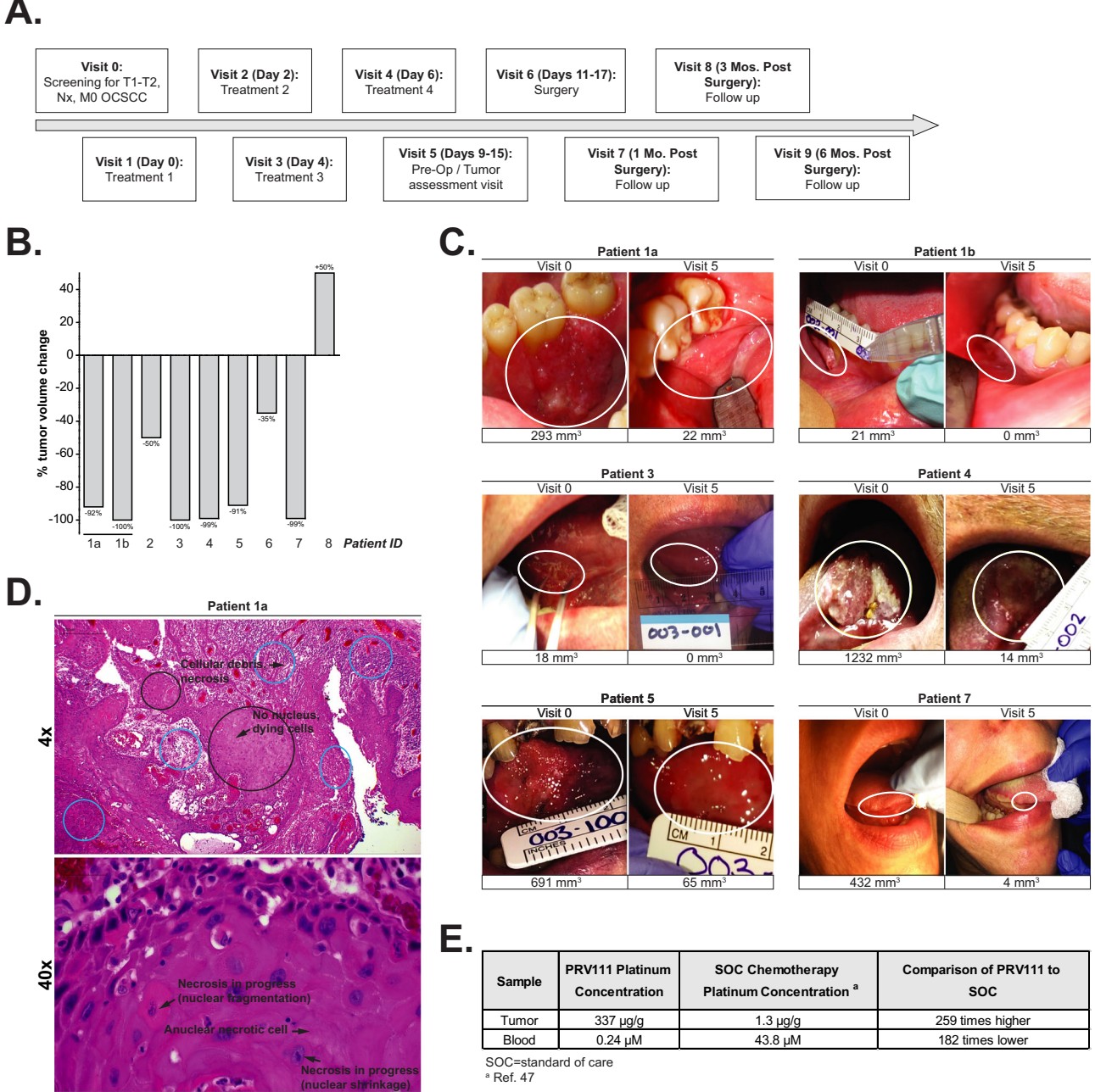

**Fig. 5 PRV111 induced a rapid anti-tumor response in patients with locally advanced OCSCC. A** Schematic representation of the clinical trial timeline. **B** Tumor volume changes from baseline and end of study for each patient treated with PRV111. **C** Tumor photographs and volume measurements for subjects which achieved >90% tumor volume reduction post RPV111 treatment. **D** Representative H&E image of post-PRV111 treatment tumor sample. *Top:* Background of residual cancer cells (black circles) and evidence of treatment effect in the form of necrosis/ulceration, cellular debris, and acute/chronic inflammation (blue circles). 4×; scale bar – 1000um. *Bottom:* Area of minimal residual cancer cells and evidence of treatment effect in the form of necrosis/ulceration and cellular debris, as indicated by black arrows. 40×; scale bar—100µm. **E** Biodistribution of platinum detected in the blood, and tumor tissue following PRV111 therapy compared with standard of care cisplatin treatment.

intravenously delivered CLPs was previously reported in several types of cancer[77,78], including syngeneic subcutaneous HNSCC model[22], only limited information is available on efficacy of localized buccal administration of nano-sized drug carriers to OCSCC.

Our group has developed PRV111, a noninvasive patch-based DDS composed of mucoadhesive polymeric matrix, embedded CLPs, and a non-permeable, pressure-sensitive, acrylic adhesive non-woven fabric backing, specifically designed for topical cis-platin delivery to mucosal tissues[17]. This study examined the safety and efficacy of PRV111 using well-characterized animal models for oral cancer and assessed the ability of neoadjuvant PRV111 to improve clinical outcome and reduce toxicity in patients with early-stage OCSCC.

A proof-of-concept experiment in murine model bearing subcutaneous tumors induced by human HNSCC cell line, demonstrated a significantly increased local retention, efficacy, and safety of locally injected CLPs, compared to non-encapsulated drug. Higher concentration of cisplatin in the tumors and lower concentration in the blood of animals treated

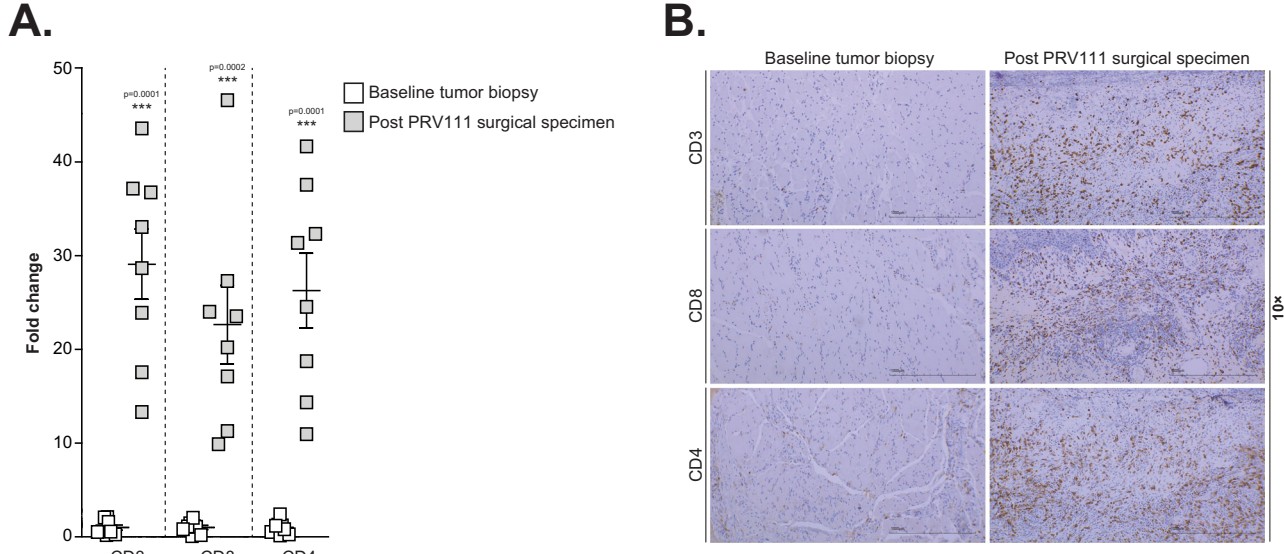

**Fig. 6 PRV111 increased the number of TILs. A** Fold change increase in TILs after treatment of 8 patients with PRV111 relative to pretreatment biopsies, shown as average ± SEM ($p < 0.001$; Student's unpaired $t$-test). Four slides were analyzed for each specimen and average was used for fold change calculation. **B** Representative images (10×; scale bar—1000μm) of pre- and post-treatment sections stained for CD3, CD8, and CD4 T cells. Changes in brown staining indicates increase in tumor-infiltrating lymphocytes following PRV111 treatment. Source data are provided as a Source Data file.

with CLPs is attributed to multiple factors, including their optimized size[17] as well as improved cell uptake and local retention due to the cationic surface charge of chitosan NPs[79]. Unlike the free-form cisplatin, which rapidly enters systemic circulation causing side effects, high concentration of CLPs remain in the tumor without permeation through the vasculature, resulting in a stronger anti-tumor response. While these results indicate that CLPs are a promising carrier for local cisplatin administration, intratumoral injection is not commonly adopted. In part due to its invasive nature, difficulty to control sustained drug release, and the development of dose-limiting toxicity in tissues surrounding the injection site[58,80]. A non-invasive PRV111 patch coupled with permeation enhancer was designed to overcome these challenges and improve the intratumoral administration efficacy by preserving CLPs in a stable environment, minimizing surface washout, enhancing the passage across epithelial barriers[17], and proving a time-controlled drug release.

An orthotopic Golden Syrian hamster buccal pouch tumor model, a well-characterized system for studying oral cancer[81,82], was chosen to assess the in vivo efficacy of PRV111. The cheek pouch is covered by a thin layer of stratified squamous epithelium that is very similar to the floor of the mouth and the ventral surface of human tongue, the most common site of OCSCC. Tumors generated by injection of HCPC-1 hamster buccal-pouch carcinoma cell line, exhibit many similarities to human disease, including morphology, histology, infiltration, metastasis, and molecular alterations[27,83–85]. Topical application of PRV111 to human sized tumors induced rapid and significant antitumor effects, with nearly full regression achieved after 4 treatment rounds. These observations, coupled with high cisplatin retention in the tumor tissue, indicate that CLPs, gradually released from PRV111, are able to penetrate deeply enough through large tumors to destroy the vast majority of cancer cells. Notably, acute nephrotoxicity induced by systemic administration of non-encapsulated drug[86], was completely abrogated in animals treated with PRV111, which may be explained by only negligible distribution of cisplatin in blood and major body organs, including the kidneys.

Patients with OCSCC can harbor multi-focal and diffuse mucosal dysplastic premalignant lesions[40]. As some of these lesions have an increased risk of malignant transformation[40,87], they represent intermediate steps in OCSCC progression[88,89]. Given OCSCC's guarded prognosis and high mortality, prompt treatment of oral premalignant lesions before they have invaded and acquired malignant potential may confer a significant survival benefit. Although surgical resection remains the main treatment for dysplastic neoplasms, oral premalignancies frequently recur and progress to malignant disease[90]. Similar to the management of premalignant cutaneous actinic keratosis[91], chemoprevention, including application of topical drugs, emerges as an alternative strategy for management of oral dysplasia. However, there is still insufficient evidence that these treatments are effective in preventing malignant transformation[90,92], posing an unmet clinical need for developing practical and noninvasive treatment modalities for precancerous lesions and early-stage malignancies that can be precisely performed. Sustained exposure of hamster buccal pouch to DMBA triggers development of oral carcinomas that are preceded by preneoplastic lesions[82], similar to those seen in humans[93], making this model ideal for testing the efficacy of PRV111 induced chemoprevention. Topical PRV111 application resulted in complete resolution of dysplastic lesions after only 3 treatments, with no overt toxicity, and significantly favorable cisplatin biodistribution profile, compared to systemic drug administration. Importantly, no recurrences were seen 35 days after the last treatment, suggesting that PRV111 may present a strategy to "intercept" cancer development, obviating the need to treat fully developed OCSCC, which acquires multiple resistance mechanisms limiting the treatment of established cancer.

A phase I/II trial initiated following encouraging in vivo results, revealed that neoadjuvant PRV111 was well tolerated by patients with early stage OCSCC (no dose-limiting or systemic toxicity, and no severe AEs) and demonstrated a remarkable technical success. Taste of the patch and the PE was reported as tolerable by all study subjects, adhesion to mucosal surfaces was persistent and complete, and uniform drug release was evident between subjects and across applications. 87.5% of the patients

showed clinical response, with 5 (62.5%) subjects demonstrating over 90% reduction in tumor volume. Paralleling biodistribution pattern observed in animal models, cisplatin concentration in tumors treated with PRV111 was many folds higher compared to the IV cisplatin administration[47,48], with only negligible amounts found in systemic circulation. A rapid response and significant local cytotoxic effects induced by PRV111 may constitute a different mechanism of action compared to the systemic chemotherapy. Interestingly, while systemically delivered cisplatin has poor penetration into the peripheral locoregional lymphatics, due to their anatomy and monodirectional flow[63], high levels of cisplatin were observed in cervical lymph nodes ablated from patients treated with PRV111. Since survival of patients with OCSCC is negatively impacted by lymph node metastasis[94], the ability of PRV111 applied at the tumor site to deliver a high concentration of the cytotoxic drug to the regional lymphatic tissue, may be paramount for treating patients with subclinical and overt tumor-draining nodes. As anticancer activity of cisplatin is broad and not limited to mitosis inhibition[52], future studies are warranted to precisely define how PRV111-induced LN cisplatin retention, and its ability to alter antitumor immunity[95], contributes to maximizing patients' long-term survival.

Taken together, this study indicates that PRV111 is an effective and safe method for delivering local cisplatin. As PRV111 mucoadhesive system can be engineered with a particular CLP release profile to tailor different clinical applications[17,65], it lays the groundwork for its potential incorporation as a clinical management tool not only for OCSCC, but also other mucosal malignancies.

## Methods

**Chemicals and reagents**. Cisplatin was purchased from Strem Chemicals (Newburyport, MA, USA). Chitosan chloride salt (Cl113) and chitosan Glutamate (G113) was obtained from Novamatrix, FMC BioPolymer (Sandvika, Norway). Sodium tripolyphosphate (TPP), glacial acetic acid and glycerol were purchased from Sigma-Aldrich (St. Louis, MO, USA). 2-dimethylaminopropanoate hydrochloride (DDAIP.HCl) was purchased from Apricus Biosciences (San Diego, California, USA). Deionized water and phosphate buffer saline (PBS) were purchased from Life Technologies (Grand Island, NY, USA)

**Cell lines**. Human HNSCC FaDu cell line was obtained from ATCC (Manassas, VA, USA), and golden Syrian hamster cheek pouch epidermoid carcinoma cell line HCPC-1 was kindly provided by Dr. Shklar (Harvard School of Dental Medicine, Boston, MA, USA)[26]. Cells were cultured in DMEM medium (ThermoFisher, Waltham, MA, USA), supplemented with 10% (v/v) fetal bovine serum, 1% penicillin-streptomycin and 1% L-glutamine (ThermoFisher) and incubated at 37 °C with 5% $CO_2$. The cells were periodically monitored for mycoplasma using the MycoDtect kit (Greiner Bio-One). All experiments were performed within 6 months of the mycoplasma screen.

**Animal models**. All animal procedures were performed in accordance with National Institutes of Health Animal Care guidelines and Massachusetts Institute of Technology, Division of Comparative Medicine requirements, under a protocol 1112-115-15 (2014) approved by the Institutional Animal Care and Usage Committee. Animals were randomized at a 1:1 male/female ratio to account for any biological variables between the sexes that may impact treatment outcome.

*Mouse subcutaneous tumor model*. Athymic nude mice (6–8 weeks old) were purchased from Charles River Laboratories (Wilmington, MA, USA) and kept in a pathogen-free facility with 12 light/12 dark cycle, 23 °C, and 40-50% humidity. FaDu cells in the logarithmic growth phase were resuspended in PBS to make a single cell suspension. 200 µl PBS solution containing $1 \times 10^7$ cells was injected subcutaneously onto the backs of mice using a 20-gauge needle. The tumors were allowed to grow for two weeks prior to treatment. Tumor-bearing mice were randomized into 5 groups (n = 6 per group) and treated with either: (i) intratumoral (IT) CDDP-NPs, (ii) free IT cisplatin (CDDP-IT), (iii) free intravenous cisplatin (IV) CDDP-IV, (iv) blank nano-particles (BLK-NPs), and (v) phosphate buffer saline (PBS) IT injection. All the mice in the drug groups received 23 µg of CDDP total, whether encapsulated in the NPs, free form IT or IV injection. In all the experiments PBS was used as a solvent. The animals were treated twice per week for 18 days. Tumor size was measured bi-weekly blinded to the treatment

group and tumor volume was calculated using the ellipsoid formula (L x W x H x π/6)[96].

*Hamster cheek pouch tumor model*. The golden Syrian hamster is a well-documented model for representing human OCSCC[81]. Hamsters (8 weeks old) were purchased from Charles River Laboratories and kept in a pathogen-free facility with 12 light/12 dark cycle, 23 °C, and 40-50% humidity. A single 200 µL injection of $1.8 \times 10^7$ HCPC-1 cells ($9 \times 10^7$ cells/mL) was administered in the left cheek pouch of hamsters using a 1 mL luer-lock syringe with a 20-gauge needle and proper sterile technique. The tumors were allowed to grow for 12 days prior to treatment. Animals were randomized into 4 groups (n = 6 per group): (i) treated with PRV111 patch containing 2 mg of cisplatin; (ii) treated with intraperitoneal (IP) injection of free CDDP solution (1 mg/kg body weight); (iii) treated with drug-free patch (BLK-patch); (iv) no treatment. Additional group contained tumor free animals without treatment. Of note, given that hamsters do not have a readily accessible tail vein, and jugular vein administration is invasive and may endure additional stress to the animals, CDDP-IP delivery was used in lieu of IV injection. Treatment was given twice per week for 17 days. Hamsters were observed for signs of toxicity, and body weight and food consumption were recorded. Tumor size was measured bi-weekly blinded to the treatment group and tumor volume was calculated using the ellipsoid formula (L x W x H x π/6). Note, since in this model PRV111 was applied topically to the human-sized tumors (hamster cheek pouch tumors can approach 1 cm in diameter, similar to small T1 tumors in humans), cisplatin dosage was based on tumor surface area ($cm^2$), rather than the body surface area (BSA) or a body weight. Therefore, a human-sized dose ($0.5 \, mg/cm^2$) was administered to the hamster tumors. There was no need to adjust the dosage based on the body weight, which is typically done for systemic administration due to dose-limiting toxicities.

*Hamster model with cheek pouch precancerous lesions*. To induce precancerous lesions, hamsters were anesthetized and the left cheek pouch was painted with 1% w/v DMBA solution (Sigma-Aldrich) in mineral oil (Millipore) five times per week for 4 weeks[29–33]. To limit local diffusion of the carcinogen, a small diameter sable brush was used for DMBA application. A subset of animals were sacrificed, biopsies of the cheek pouch lesions were removed for histological examination, and generation of dysplastic lesions was confirmed by the senior veterinary pathologist. The remaining animals were randomized into 3 groups (n = 3 per group): (i) treated with PRV111 patch containing 2 mg of cisplatin; (ii) treated with IP injection of free CDDP solution (1 mg/kg body weight); (iii) treated with drug free BLK-patch. Treatment was given twice per week for two weeks. Animals were periodically examined (for 49 days) to record the presence of dysplastic lesions by palpation of the pouch surface and visually assessing for leukoplakia at 2X magnification. All lesions were evaluated by a senior veterinary pathologist.

The maximal tumor size permitted in nude mice (≤1.5 cm longest dimension) and hamsters (≤2 cm longest dimension) was not exceeded. Weight and overall health of the animals (such as food and water intake, abnormal stool/urine, moist fur or mouth, rapid or labored breathing, physical appearance and behavior) as well as local toxicity were monitored daily. Animals were euthanized by $CO_2$ inhalation at the endpoint, or when signs of excessive tumor burden, illness or distress were present.

**Nanoparticles preparation**. Optimization was performed by altering the mass ratio of chitosan to TPP and synthesis *p*H in order to define a formulation which could yield NPs of desirable size and charge, with low polydispersity and high encapsulation efficiency[17]. CDDP-NPs and BLK-NPs were synthesized using the ionic gelation technique. Briefly, chitosan Cl113 was hydrated in 0.175% acetic acid (1.0 mg/mL) and stirred until completely dissolved. TPP (cross linker used for ionic gelation) was dissolved in water (0.45 mg/mL) and filtered through a 0.22 µm syringe filter. CDDP (1.0 mg/mL) was added to this solution and the mixture was sonicated until CDDP was completely dissolved. The CDDP-TPP solution was added drop-wise at a constant flow rate, resulting in a final volume ratio of 1.1 mL chitosan: 0.85 mL CDDP-TPP. For the synthesis of BLK-NPs, the TPP solution without the CDDP was combined with the Cl113 solution in a final volume ratio of 4.0 mL chitosan: 1.0 mL TPP. Note, upon sonication, CDDP undergoes hydrolysis and loses its chloride, which can result in the production of hydrochloric acid, subsequently lowering the pH. The lower pH level in the CDDP-NP synthesis (compared to the synthesis of BLK-NP) requires an adjusted amount of TPP as a cross linker.

**Preparation of PRV111 patch**. Briefly, CS G113 was hydrated in a 1.0% v/v aqueous acetic acid solution (17.0 mg/mL) under magnetic stirring for one hour at 700 rpm. Permeation enhancer DDAIP.HCl (50 mg/mL) and glycerol (50% w/w of CS G113) were added and the solution was stirred for 30 min before being added to either CDDP-NP or BLK-NP solution in a ratio of 10.0 mL: 1.0 mL. Taste masking [peppermint (0.57% v/v) and EZ-Sweetz (0.28% v/v)] was added to the solution and gently mixed. The mixture was frozen in liquid nitrogen for 25 min and lyophilized for two days. For clinical trial, the PRV111 patch and PE powder for reconstitution was packaged in a self-sealing pouch and enclosed in a secondary

heat sealed, triple laminated, Mylar foil pouch to protect the patch from light and moisture. The lot numbers of PRV111 and PE were recorded during the study.

**Patch application for in vivo experiments.** Hamsters were anesthetized during the treatment with 1-4% isoflurane in an induction chamber. Once anesthetized, the hamster's cheek pouch was pulled out for access and permeation enhancer was brushed topically using a fine-tipped brush applicator. After 15 min, the area was cleaned and a PRV111 or BLK-patch was applied topically on the lesion surface and kept for one hour. Moist gauze was placed on the exposed backing side of the patch to keep it hydrated during the treatment period to allow proper drug release. The hamsters remained anesthetized with a constant flow of isoflurane and were kept warm with a heating pad during PRV111 treatment.

**In vivo permeation assessment.** Orthotopic oral tumors were pretreated with PE for 5 min, and PRV111 patch loaded with Cy5 conjugated chitosan nanoparticles that encapsulated FITC was subsequently applied for 10 min. Tumors were harvested, washed with saline to remove any excess material that did not permeate, sectioned using a cryotome, and images were taken using EVOS FLoid imaging systems (ThermoFisher). For experiments using normal porcine buccal tissue (purchased from Lemay & Sons, Goffstown, NH), specimens were collected immediately after animal was sacrificed. The remaining steps were identical to those used to treat oral tumors in a living hamster. Due to its close resemblance to human buccal mucosa in structure, enzyme activity and permeability characteristics, porcine buccal mucosa has been extensively used as a model to study the permeability of various diffusants, and to assess their potential to be delivered through the buccal route[38,39]. Since human oral mucosa is not widely available, normal porcine buccal mucosa was used to assess permeation of nanoparticles loaded to PRV111.

**Histopathological examination for in vivo experiments.** At the endpoint of animal experiments, organs including kidneys, lungs, liver and spleen were harvested and gross necropsy was performed. Samples were fixed in 10% neutral buffered formalin, processed, paraffin embedded, sectioned, and stained with Hematoxylin and Eosin (H&E) at MIT Koch Cancer Institute pathology core. Slides were examined by Roderick Bronson, D.V.M. for pathological changes.

**Platinum quantification for in vivo experiments.** Quantification of platinum (Pt) in blood (pharmacokinetic analysis) as well as in tumor samples and organ tissues (biodistribution analysis) was performed by Excite Pharma Services LLC (Lee's Summit, MO, USA) in their FDA inspected facility which adheres to Good Manufacturing Practices (GMP) and Good Laboratory Practices (GLP). The analysis was performed using Perkin Elmer Nexion 350X inductively coupled plasma mass spectrometry (ICP-MS) instrument (Waltham, MA, USA) and a validated (ICP-MS) method[34–37]. Lower limit of quantitation [LLOQ] = 1.06 ng/mL.

**Statistical analysis for in vivo experiments.** All data are presented as the mean ± SEM. The Wilcoxon Rank-sum method was used to analyze the differences between groups with values not evenly distributed across populations. This method was selected for the analysis of in vivo studies due to the large disparities in tumor volumes between the PRV111 and non-PRV111 treated groups. This method was accepted and agreed upon by FDA. Student's unpaired t-test (two-sided) was used to analyze differences between two groups. When appropriate, the Bonferroni correction was applied to account for multiple comparisons. Results with $p < 0.05$ were considered significant. Statistical analyses were conducted using GraphPad Software (version is 9.4.0). P-values are summarized as: *$P ≤ 0.05$, **$P ≤ 0.01$, and ***$P ≤ 0.001$, unless stated otherwise. All representative images reflect a minimum of three biological replicates.

**Clinical study eligibility.** The CLN-001 study is an open label, single-arm trial of PRV111 in resectable early-stage oral cavity cancer. Key eligibility criteria included pathologically and clinically confirmed T1 (≤2 cm) or T2 (≥2 cm but ≤4 cm) SCC of the mucosal lip or oral cavity amenable to surgical resection. Previous radiation for head and neck cancer was excluded. Additional criteria included ECOG performance status of ≤2, adequate renal function, and absence of serious underlying medical conditions which could impair the ability of the subject to participate in the study. Patients were enrolled between August 30th 2018 and May 6th 2020. Detailed description of the inclusion/exclusion criteria is provided in the clinical trial protocol submitted as Supplementary Note 1 in the Supplementary Information file. Trial was approved by IRB, registered on clinicaltrials.gov (NCT03502148; https://clinicaltrials.gov/ct2/show/NCT03502148), and complies with the ICMJE guidelines on reporting.

**Safety and evaluable populations.** Subjects who have at least 1 PRV111 patch applied were to be included in the safety population (n = 10). The safety population was used for all summaries of safety data. Subjects who have met the inclusion and exclusion criteria, and completed 4 treatments with PRV111 (n = 8) were included in the evaluable population. This population was used for the efficacy analysis and assessed for tumor response.

**Study design.** This is a 2-Stage adaptive study, where Stage 1 focuses on determining a safe dose, while Stage 2 expands the study at the previously determined dose[43–45]. For Stage 1, ten subjects were enrolled for the purpose of 5 evaluable subjects for determining a safe and efficacious dose. For Stage 2, the target goal was up to 11 evaluable subjects for the efficacy analysis of the final dose. Notes: (a) As all safety and efficacy endpoints were met for Stage 1, the dose was not changed for Stage 2; (b) As number of responses required at the final dose for the trial to be deemed successful was >6 (see Power and Sample Size section below), once 7 responses were reached, the primary endpoint was met, and recruitment was stopped at 8 evaluable subjects. Detailed description of the study design is provided in the clinical trial protocol submitted as Supplementary Note 1 in the Supplementary Information file.

**Power and sample size.** The planned enrollment for the final dose is intended to obtain up to $N = 11$ evaluable subjects. Based on Simon's 2-stage procedure[45,97], the null hypothesis that the true response rate is 0.30 (where a response is the occurrence of complete response, strong response or partial response - see response/non-response criteria in statistical analysis plan submitted as Supplementary Note 1 in the Supplementary Information file) will be tested against a 1-sided alternative. In the first stage, 5 evaluable subjects will be accrued. If there are 2 or fewer responses in these 5 subjects, the dose will be escalated. Otherwise, the dose will remain the same or de-escalated based on safety data. Additional subjects will be accrued for a total of up to 11 evaluable subjects at the final dose. The null hypothesis will be rejected if 7 or more responses are observed at the final dose. This design yields a type I error rate of 0.018 and power of 96.44% when the true response rate is 0.85 (Supplementary Table 1). Note, the null hypothesis was rejected due to having 7 responses in the final dose in the initial 8 evaluable subjects and the study was completed early.

**Clinical study treatment.** Three weeks prior to surgery, patients received one cycle of PRV111, consisting of 4 treatment visits (separated by at least 2 days but not more than 6 days). During each visit, one or two 4 cm² PRV111 patches (depending on tumor size) were applied to the tumor area for 5 min after PE application. Three successive 10-minute patch applications were performed. Each visit dose: ≤12 mg of cisplatin, each patch loading dose: 2 mg of cisplatin. PE powder was reconstituted in 1 mL US Pharmacopeia Water prior to application.

**Tumor measurements.** PRV111 is a chemoablative therapy that works by destroying tumors from the top down. As traditional methods of determining systemic chemotherapy response do not work well in the topical model, for this study, volumetric tumor measurements were accepted by FDA as the primary method of response. Tumors were measured in cm (to the tenth decimal place) using a disposable plastic ruler or caliper. Three dimensions were recorded (length x width x height). If the tumor was flat, 0.05 cm was recorded for the height. If the tumor was concave, a negative height was recorded. If the height was an unmeasurable dent, it was recorded as −0.05 cm. The following formula was used to calculate tumor volume (cm3): L x W x H x π/6. Response was defined as percent tumor volume change from the baseline. Detailed description of the response criteria is provided in the clinical trial protocol submitted as Supplementary Note 1 in the Supplementary Information file.

**Pharmacokinetic assessments in patients' samples.** Whole blood samples were collected at −15 to 0 min prior to administration of PE and PRV111 application, and at 30 ± 5 min, 60 ± 5 min and 120 ± 5 min following application of the last patch at Visit 1 (first treatment visit) and at Visit 4 (last treatment visit). At Visit 2 and Visit 3, whole blood samples were collected at −15 to 0 min prior to administration of PE and PRV111 application, and at 30 ± 5 min after application of the last patch. Platinum content in blood samples as well as in tumor and lymph node specimens obtained following surgery was assessed at Excite Pharma Services LLC as described above.

**Quantification of tumor-infiltrating lymphocytes (TILs).** Serial freshly cut 4-μm sections of the blocks containing pre-treatment biopsy or post-treatment surgical specimens were stained for anti-CD3 (1:100; Ventana, 790-4341), anti-CD8 (1:100; Ventana, 790–4460) and anti-CD4 (1:100; Ventana, 790-4423) using Ventana Benchmark automated method at the Dana Farber Cancer Institute, Specialized Histopathology Service Core. Slides were digitalized using Aperio (Leica Biosystems) scanner and number of lymphocytes that immunostained positively was counted using ImageJ software (NIH).

**Institutional review board (IRB) and informed consent.** This study was approved by the IRB representing the participating institution. Written informed consent was obtained from each subject prior to performing any study-related procedures. Study participants provided informed consent for publication of the tumor images in Fig. 5. The trial was designed and monitored in accordance with the Good Clinical Practice (GCP) principles and the Declaration of Helsinki.

**Reporting summary**. Further information on research design is available in the Nature Research Reporting Summary linked to this article.

## Data availability

The authors declare that the data supporting the findings of this study are available within the article and its supplementary information files. Source data are provided with this paper. The clinical study protocol and statistical analysis plan are provided as Supplementary Note 1 in the Supplementary Information file. Patients' de-identified data (such as diagnosis, gender, averaged age, and aggregated ethnicity) is provided in the manuscript.

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

## Acknowledgements

This work was supported by the NIH grants R43 DE023725 and R44 CA192875, FDA Office of Orphan Products Development grant R01 FD006325, and National Science Foundation Award NSF IIP-1315084. E.I. effort was supported by the NIH grant R01DE027809. E.A. salary for partially supported by the TUBITAK 2219 postdoctoral research scholarship.

## Author contributions

Conceptualization: M.G., A.M., N.A., and E.I. Methodology: M.G., A.M., N.A., E.I. S.C., P.C., and B.F. Validation: M.G., A.M., V.M., A.S. S.C., P.C., B.F. A.B., B.L., S.Y., V.S., J.M., and J.S. Investigation: M.G., A.M., S.C., P.C., B.F. E.R.G., A.V., S.E.O., E.A., M.J.A., and S.G. Resources: M.G., and A.M. Data curation: A.M., M.G., E.I., N.A., R.H., M.L., P.C., S.C., and B.F. Writing - original draft preparation: E.I., A.M., M.G., N.A., S.C., V.M., and A.S. Writing - review and editing: E.I., M.G., A.M., S.C., V.M., A.S., N.A., A.J.R., A.Z., A.T.P., N.L., M.L., R.H., G.G., C.B., P.C., B.F and A.D. Visualization: E.I., A.M., V.M., A.S., and M.G. Supervision: M.G., A.M., N.A., and E.I. Funding acquisition: M.G.

## Competing interests

M.G., A.M., A.B., B.L., P.C., S.C., B.F. and E.R.G. are affiliated with Privo Technologies. N.A. and S.G. serve as advisors for Privo Technologies. The remaining authors declare no competing interests.

## Additional information

[1]David H. Koch Institute for Integrative Cancer Research, Massachusetts Institute of Technology, Cambridge, MA, USA. [2]Harvard-MIT Division of Health Sciences and Technology, Massachusetts Institute of Technology, Cambridge, MA, USA. [3]Department of Biomedical Engineering, University of Massachusetts Lowell, Lowell, MA, USA. [4]Privo Technologies, Peabody, MA, USA. [5]Department of Medicine, Section of Hematology and Oncology, University of Chicago, Chicago, IL, USA. [6]Department of Surgery, Section of Otolaryngology-Head and Neck Surgery, University of Chicago, Chicago, IL, USA. [7]Department of Otolaryngology-Head and Neck Surgery, Johns Hopkins University School of Medicine, Baltimore, MD, USA. [8]Department of Neurosurgery, Johns Hopkins University School of Medicine, Baltimore, MD, USA. [9]Department of Neurosurgery and Oncology, Johns Hopkins University School of Medicine, Baltimore, MD, USA. [10]Department of Oral Maxillofacial Surgery, The University of Texas Health Science Center at Houston, Houston, TX, USA. [11]Department of Otolaryngology-Head & Neck Surgery, Baylor College of Medicine, Houston, TX, USA. [12]Massachusetts College of Pharmacy and Health Sciences, Boston, MA, USA. [13]Insilico Medicine, Pak Shek Kok, Hong Kong. [14]Department of Pharmacy and Pharmaceutical Technology, University of Santiago de Compostela, Galicia, Spain. [15]Department of Pathology, University of Chicago, Chicago, IL, USA. [16]Department of Oncology, Apollo Hospital, Mumbai, India. [17]These authors contributed equally: Manijeh Goldberg, Aaron Manzi. ✉email: mgoldberg@privotechnologies.com; nagrawal@surgery.bsd.uchicago.edu; izumchen@uchicago.edu

