## [Peer Review File · Nature Communications]

A nanoengineered topical transmucosal cisplatin delivery system induces anti-tumor response in animal models and patients with oral cancerREVIEWER COMMENTS

Reviewer #1 (Remarks to the Author): with expertise in nanoparticles, patches

In this manuscript, Goldberg et al. reported the safety of PRV111 patch in animal models and patients. The study is well designed and performed. The authors successfully validated the antitumor efficacy of the cisplatin-loaded chitosan patch against oral cavity squamous cell carcinoma. Thus, I enthusiastically support acceptance of this work after addressing the following minor issues.

- 1) A brief characterization of nanoparticles and PRV111 will be informative.
- 2) Could the authors mention the therapeutic efficiency without the application of PE ?
- 3) More discussions about cisplatin-associated toxicities are beneficial in the introduction.
- 4) The photo or the description of how the PRV111 was applied to the mouse cavity may be helpful for readers to understand this procedure.
- 5) Error bars are missing in the CDDP-NP group in Fig.1D.

Reviewer #2 (Remarks to the Author): with expertise in oral cancer, clinical

Systemic platinum-based chemotherapy can often lead to significant toxicity in oral cancer patients. In this study, the authors developed PRV111, a noninvasive patch-based drug delivery system specifically designed for topical delivery of cisplatin-loaded chitosan particles (CLPs) to oral cancers. To test if PRV111 application could penetrate tumor tissue and avoid systemic circulation and toxicity, three animal models (mouse and hamster) of oral cancer were used, demonstrating the superior local retention, efficacy, and safety of CLPs using PRV111 as compared to either non-encapsulated or intravenous cisplatin administration. To further assess the efficacy and safety, a pilot phase I/II clinical trial was conducted in 10 patients with resectable OCSCC by neoadjuvant local administration of PRV111. PRV111 treatment rendered 69% tumor reduction in ~7 days and over 87% response rate in 8 patients. Significantly lower cisplatin was found in patients' blood compared to their tumor. No dose-limiting toxicities, serious adverse event, or systemic toxicities were observed. TILs were also remarkably increased after PRV111 treatment.

Both animal models and clinical trial demonstrated PRV111's clinical activity and low toxicity profile. Their data supports that local treatment with PRV111 may provide a safer and more effective alternative to systemic cisplatin, potentially improving outcomes and quality of life of oral cavity cancer patients. The application in oral pre-malignant lesions at high risk of transformation is very interesting and a major unmet medical need. The manuscript describes a novel cisplatin-delivery system, has compelling safety and efficacy data and direct clinical application. It deserves to be accepted for publication.

Major comment:

1. Volumetric assessment can overestimate response and is not a validated prognostic factor. Please report on the pathologic response/percentage of viable tumor cells in the surgical specimen of the trial patients.

Minor comments:

2. Fig 1B, no Standard Deviation (SD) bar was shown on CDDP-IT except the last timepoint and no SD on CDDP-IV. Is the deviation too small to show on the figure? Usually the tumor size from each mouse has certain variation.
3. In Fig 2, PRV111, which embedded with encapsulated cisplatin, was compared with CDDP (non-encapsulated). Have you compared CDDP-NP (nanoparticle form) with PRV111 in animal model? Per Fig 1, CDDP-NP suppressed tumor much more effectively compared to CDDP via IT (intra-tumor

injection). Fig2E, was the adjacent normal tissue checked? any uptake of chitosan particles and any toxicity on the adjacent normal tissue?

4. Fig 4B (DMBA-induced tumor model in hamster), were multi-lesions induced by DMBA in each animal? 3 animals were used for each group and no statistical analyses were included.

5. Line 519-- "with the pretreatment biopsies (Error! Reference source not found., 6B)", need to correct reference citation here.

6. Please provide T stage of patients enrolled in the trial

Reviewer #3:

Editorial note – please note that Reviewer #3 (with expertise in biostatistics and clinical trial study design) did not provide any Remarks to the Author. However, while the referee highlights that the study design is appropriate, they have confidentially raised a concern on the very few sample size and the very large effect size defined (85% vs. 30% in response rate). Please explain why a smaller effect size (e.g., 60% vs. 30%) was not chosen.

Reviewer #4 (Remarks to the Author): with expertise in oral cancer, animal models

In this paper, the authors demonstrated the efficacy and safety of a patch system to deliver cisplatin. In different animal models, they showed that the patch was therapeutically useful to treat tumors, inhibit recurrences and the development of premalignant lesions, with no toxicity. They also performed a clinical trial in HPV negative patients with positive results. Their findings will help to improve cisplatin delivery and will enhance other therapies in combination with this system.

I have some comments to the authors that I would like them to address:

-in the introduction section:

*clarify the first time you use IV writing intravenously (IV) as you did with other abbreviations.

-in M&M:

*in chemicals and reagents, DMBA and mineral oil are lacking. They are mentioned in the premalignant lesion animal model paragraph but I think they have to be mentioned with the other reagents.

*"All animal procedures were performed in accordance with a protocol approved by the Institutional Animal Care and Usage Committee": I think this statement should be located before the description of all the procedures made in the animals. Also, it is important to mention the number and date of approval of the animal protocols presented and the animal welfare guidelines used to perform these experiments. Particularly, I was wondering the final humane endpoint used related to tumor volume. That is important related to the animal's welfare, as the authors worked with "human-sized tumors".

*How biopsies of premalignant lesions were performed to confirm dysplasia?

*A comment: Why the authors anesthetize the animals during their treatment with DMBA? I would like to share with the authors that it is possible to apply DMBA topically with a syringe (without needle) with the animal awake. The procedure is quick and the animal does not experience discomfort.

*In hamster, intravenous injection can be performed under anesthesia making a small incision in the skin of the neck, exposing the jugular vein. In this sense the sentence "Of note, due to the inability to perform tail vein CDDP-IV administration in hamsters, CDDP-IP delivery was used instead of the IV injection" should be removed.

*Please, unify how many applications were performed (5 or 6?), in the following paragraph: "Hamsters were treated for 17 days (6 treatments total). Animals treated with PRV111 demonstrated substantial reduction in tumor sizes after just 3 treatments (day 10) (Figure 2B), with complete tumor regression achieved in most animals by day 17 (after 5 treatments with PRV111)"

*In the hamster model with precancerous lesions, when it says "Biopsies of lesions were collected from untreated animals for histological confirmation of dysplasia" corresponds to the control group, cancerized with no treatment?

Please correct "CO2".

How the authors assessed toxicity after treatment? which parameters they evaluated? for example, mucositis development?.

*In vivo permeation study was performed in porcine tissue, the animals should be mentioned in the text and a description should be added. Comment #1: Why did the authors study in vivo permeation in porcine tissue and not in the hamster normal pouch? Comment #2: Why the authors did not assess the permeation treatment in oral precancerous lesions?

Results:

*There are no control groups, ie, animals with tumors without any treatment. Those models are well reported in previous studies? For the case of the oral precancerous lesion model, the cited bibliography does not match exactly with the cancerization protocol, have you characterized this model previously?.

*In the text, it says, "Animals in the 356 CDDP-IV group were euthanized at day 7 after treatment initiation due to the rapidly decreasing body weight and signs of toxicity associated with high level of systemic CDDP circulation (Figure 1B)". It has to say, "figure 1E"

*In Supplementary Figure 1 legend, it says: "...Immunodeficient nude mice bearing subcutaneous human HNSCC FaDu cell line induced tumors treated with 878 IP administration of CDDP-NPs". If I understood correctly, no IP injection was performed in this model.

*In figure 2E, the authors should explain in the figure legend what "yellow" fluorescent in the photo means.

*In the paragraph "PRV111 increased the number of tumor-infiltrating lymphocytes" line 519, correct the error.

*Could the authors clarify the following paragraph?:

"Topical administration of PRV111 reduced cisplatin associated toxicities", it says that "Analysis performed one hour after administration of the first dose demonstrated that blood level of cisplatin in the CDDP-IP group was over 20-fold higher than the level detected in PRV111 treated animals (Figure 3A). Hamsters were sacrificed and major organs such as liver, lungs, kidneys, spleen, brain and heart, as well as the tongue, healthy contralateral cheek pouch, and tumors, were isolated to evaluate the tissue distribution of cisplatin in PRV111 and CDDP-IP treated groups using ICP-MS. Note, we were able to collect very small residual tumors samples from three animals treated with PRV111, as the remaining hamsters were tumor free at the end point".

Which endpoint the authors considered? It is not clear which tumors treated with PRV111 were evaluated as in the long-term recurrence study none of the animals developed recurrences so no tumors were available.

*Figure 5 supplementary: Is it possible for the authors to provide a better photo representing a precancerous lesion in the hamster cheek pouch?

Discussion:

*Comment: The fact that PRV111 was effective in treating HPV negative tumors is important, as it is well known that HPV positive tumors respond better to radiotherapy than HPV negative tumors. In this sense, do the authors think possible to combine this strategy with radiation therapy to enhance HPV negative tumor response to radiation?

As requested, detailed responses to the comments of the reviewers are recorded below and we have revised the manuscript accordingly. We believe that the additional information provided in response to the reviewer's comments have strengthened the manuscript and we greatly appreciate their input.

REVIEWER COMMENTS

Reviewer #1 (Remarks to the Author): with expertise in nanoparticles, patches

In this manuscript, Goldberg et al. reported the safety of PRV111 patch in animal models and patients. The study is well designed and performed. The authors successfully validated the antitumor efficacy of the cisplatin-loaded chitosan patch against oral cavity squamous cell carcinoma. Thus, I enthusiastically support acceptance of this work after addressing the following minor issues.

We thank the reviewer for their valuable and thoughtful comments.

1) A brief characterization of nanoparticles and PRV111 will be informative.

Response: A brief description of nanoparticles and PRV111 patch is provided in the introduction section. Page 2: *“Building on this previous work by our group and others^{17,22}, we have developed PRV111, a self-adhesive cisplatin transmucosal system designed to deliver cisplatin-loaded chitosan particles (CLPs) to anatomically accessible oral cancers, such as lip, tongue, gum, floor of mouth, gingiva, buccal mucosa, and other locations in the oral cavity. PRV111 is a thin, 2-layer, matrix-type, polymeric transmucosal patch, consisting of a chitosan matrix layer embedded with CLPs and an impermeable ethyl-cellulose adhesive backing, designed to provide targeted drug delivery and prevent CLP washout from saliva. Each PRV111 topical patch contains 0.5 mg/cm² (2 mg) of cisplatin, and covers a tumor region of 4 cm². The system also incorporates a permeation enhancer (PE), which functions by reversibly opening the tight junctions between the cells, allowing for optimal penetration and absorption of the CLPs released from the patch. The released CLPs swell to approximately 0.5 micron when exposed to moisture, allowing them to diffuse across the porous matrix and into the tumor tissue. These particles are too large to penetrate into the vasculature (vasculature pore size is 2–15 nm), and therefore there is no systemic cisplatin exposure²⁴.”* Additionally, a detailed characterization of nanoparticles and PRV111 patch is provided in the Materials and Methods section (Page 5), under subsections titled “Nanoparticles preparation” and “Preparation of PRV111 patch”.

2) Could the authors mention the therapeutic efficiency without the application of PE?

Response: Nanoparticles loaded to PRV111 were designed to penetrate and absorb only in areas where PE (a critical component of the PRV111 system) is applied, thus minimizing the damage to the adjacent non-tumorous tissue (as described on Page 8). CLPs do not permeate if PE is not applied prior to patch administration (**Supplementary Figure 4**). If the patch was to be applied on the surface with no epithelium (i.e. resected tumor bed) than the PE would not be necessary. The thick epithelium and mucosa of the oral cavity is designed to keep things out from penetrating into the tissue. So for topical applications of PRV111, administration of PE is critical.

3) More discussions about cisplatin-associated toxicities are beneficial in the introduction.

Response: In response to Reviewer’s request, the revised version of the manuscript now contains additional text and references. Page 3: *“Surgery may be followed with adjuvant chemoradiotherapy, with cisplatin (cis-diamminedichloroplatinum(II), CDDP) being the most frequently administered systemic chemotherapeutic agent^{8,9}. Intracellularly, cisplatin acts primarily by producing DNA inter- and intra-strand crosslinks, which prevent DNA replication and promote apoptosis of tumor cells¹⁰. However, given the nonselective targeting of both healthy and malignant tissues, its clinical utility is hindered by adverse effects and toxicities, including*

protracted nausea, vomiting, ototoxicity, acute nephrotoxicity, myelosuppression, and chronic neurotoxicity^{11,12}. These side effects of systemic cisplatin administration can be dose-limiting, reducing the cumulative dosage and length of treatment for patients, and subsequently limiting therapeutic benefits¹³. Thus, development of novel therapeutic delivery approaches which enhance efficacy and reduce toxicity is paramount to improve health outcomes and survival of patients with OCSCC.”

4) The photo or the description of how the PRV111 was applied to the mouth cavity may be helpful for readers to understand this procedure.

Response: The paragraph titled “Patch application for in vivo experiments” in the Methods section provides a detailed description of how the PRV111 patch was applied. Page 5: “Hamsters were anesthetized during the treatment with 1-4% isoflurane in an induction chamber. Once anesthetized, the hamster’s cheek pouch was pulled out for access and permeation enhancer was brushed topically using a fine-tipped brush applicator. After 15 minutes, the area was cleaned and a PRV111 or BLK-patch was applied topically on the lesion surface and kept for one hour. Moist gauze was placed on the exposed backing side of the patch to keep it hydrated during the treatment period to allow proper drug release. The hamsters remained anesthetized with a constant flow of isoflurane and were kept warm with a heating pad during PRV111 treatment.” Additionally, **Supplementary Figure 3** and **Figure 2C** provide images of hamsters under anesthesia with PRV111 applied topically on the lesion surface.

5) Error bars are missing in the CDDP-NP group in Fig.1D.

Response: We thank the Reviewer for his/her notice. In fact, the error bars in the CDDP-NP group were present, but since the SD values in this group are small, the error bars were masked by the symbol (green down pointed triangles). We have reduced the size of the symbols in the revised Figure so that the error bars will be visible.

Reviewer #2 (Remarks to the Author): with expertise in oral cancer, clinical

Systemic platinum-based chemotherapy can often lead to significant toxicity in oral cancer patients. In this study, the authors developed PRV111, a noninvasive patch-based drug delivery system specifically designed for topical delivery of cisplatin-loaded chitosan particles (CLPs) to oral cancers. To test if PRV111 application could penetrate tumor tissue and avoid systemic circulation and toxicity, three animal models (mouse and hamster) of oral cancer were used, demonstrating the superior local retention, efficacy, and safety of CLPs using PRV111 as compared to either non-encapsulated or intravenous cisplatin administration. To further assess the efficacy and safety, a pilot phase I/II clinical trial was conducted in 10 patients with resectable OCSCC by neoadjuvant local administration of PRV111. PRV111 treatment rendered 69% tumor reduction in ~7 days and over 87% response rate in 8 patients. Significantly lower cisplatin was found in patients’ blood compared to their tumor. No dose-limiting toxicities, serious adverse event, or systemic toxicities were observed. TILs were also remarkably increased after PRV111 treatment. Both animal models and clinical trial demonstrated PRV111’s clinical activity and low toxicity profile. Their data supports that local treatment with PRV111 may provide a safer and more effective alternative to systemic cisplatin, potentially improving outcomes and quality of life of oral cavity cancer patients. The application in oral pre-malignant lesions at high risk of transformation is very interesting and a major unmet medical need. The manuscript describes a novel cisplatin-delivery system, has compelling safety and efficacy data and direct clinical application. It deserves to be accepted for publication.

We thank the reviewer for his/her appreciation of our study and valuable comments.

Major comment:

1. Volumetric assessment can overestimate response and is not a validated prognostic factor. Please report on the pathologic response/percentage of viable tumor cells in the surgical specimen of the trial patients.

Response: We appreciate and understand the Reviewer’s concern. PRV111 is a chemoablative therapy that works by destroying tumors from the top down. Traditional methods of determining systemic chemotherapy response do not work well in the topical model since the cell destruction pattern is different. Pathological assessment as the primary endpoint is inappropriate because several layers of necrosed cells are sloughed off during treatment, and thus the evaluation would not be representative of therapeutic effect.

This rationale was presented to FDA during the clinical trial, and the Agency agreed that volumetric assessment was the most appropriate measure for efficacy of PRV111 in this study. Because this was a first-

in-man phase 1/2 study, the goal was simply to demonstrate the safety and clinical activity of PRV111. For the pivotal trial to support registration, a validated prognostic endpoint such as disease-free survival or overall survival will be used. Privo requested for FDA to determine if the volumetric in the case of topical chemotherapy is appropriate and the Agency agreed that it is a better representative of the response to PRV111 treatment.

Page 7: *“PRV111 is a chemoablative therapy that works by destroying tumors from the top down. As traditional methods of determining systemic chemotherapy response do not work well in the topical model, for this study, volumetric tumor measurements were accepted by FDA as the primary method of response.”*

Minor comments:

2. Fig 1B, no Standard Deviation (SD) bar was shown on CDDP-IT except the last timepoint and no SD on CDDP-IV. Is the deviation too small to show on the figure? Usually the tumor size from each mouse has certain variation.

Response: We thank the Reviewer for his/her notice. As correctly suggested by the Reviewer, the SD values in these groups and time-points are small, and were masked by the symbols. We have reduced the size of the symbols in the revised Figure so that the error bars will be visible.

3. In Fig 2, PRV111, which embedded with encapsulated cisplatin, was compared with CDDP (non-encapsulated). Have you compared CDDP-NP (nanoparticle form) with PRV111 in animal model? Per Fig 1, CDDP-NP suppressed tumor much more effectively compared to CDDP via IT (intra-tumor injection).

Response: A direct comparison of response patterns between subcutaneous FaDu tumor growing in immunocompromised athymic nude mice (**Figure 1B**) and orthotopic HPLC-1 oral tumor growing in immunocompetent hamster model (**Figure 2B**) should be addressed with caution, as doubling time, niche, microenvironment and drug absorbance of these tumors are substantially different. Intratumoral delivery of CDDP-NP in murine subcutaneous tumors was performed as a proof-of-concept experiment to assess local retention, efficacy, and safety of locally injected CLPs, compared to non-encapsulated drug. As intratumoral injection is not commonly adopted in clinical setting, we didn't directly compare it with the PRV111 patch. Nevertheless, both locally injected nanoparticles and CLPs gradually released from PRV111 are able to retain cisplatin at the delivery site and induce robust anti-tumor effect.

Fig2E, was the adjacent normal tissue checked? any uptake of chitosan particles and any toxicity on the adjacent normal tissue?

Response: Nanoparticles loaded to PRV111 were designed to penetrate and absorb only in areas where permeation enhancer (a critical component of the PRV111 system) is applied, thus minimizing the damage to the adjacent non-tumorous tissue (as described on Page 8). As indicated in **Supplementary Figure 4**, CLPs do not permeate to the normal buccal tissue if permeation enhancer (PE) is not applied prior to patch administration.

4. Fig 4B (DMBA-induced tumor model in hamster), were multi-lesions induced by DMBA in each animal?

Response: While DMBA may occasionally induce multifocal dysplastic lesions at longer exposure (6-7 weeks), in our experiments all lesions were disused unifocal. To limit local diffusion of the carcinogen, a small diameter sable brush was used for DMBA application. We had added this information to the revised version of the manuscript. Page 5: *“To limit local diffusion of the carcinogen, a small diameter sable brush was used for DMBA application.”* Page 9: *“While BLK-patch treated animals displayed unifocal oral dysplastic lesions throughout the entire monitoring period, all hamsters in PRV111 group were lesion-free after 3 treatments (day 14).”*

Three animals were used for each group and no statistical analyses were included.

Response: Given that the measurer in this experiment was either presence or absence of the lesion, lesion related statistics were observational. Although the Reviewer #3 of this manuscript (an expert in biostatistics) considered this work adequately supported, we are providing the weight related and lesion related statistics per Reviewer's request.

Weight-related Statistics

H0: No statistically significant weight difference between the 2 groups. (Null hypothesis was False). There was no statistically significant difference between PRV111 as compared with the IP Cisplatin or + Control groups

Weight Statistics

Comparison	P Value	Statistically Significant (<0.05)
PRV111 vs CDDP-IP	0.29	*No (no significant weight difference)
PRV111 vs Control	0.16	*No (no significant weight difference)
CDDP-IP vs Control	0.48	*No (no significant weight difference)

Abbreviations: IP=intraperitoneal; Wt=weight

This observation is reported in the manuscript. Page 9: “.....the body weights were not significantly different between the groups (Figure 4C).....”

Lesion Related Statistics

- PRV111:
 - (Day 14) 100 % of all hamsters in this group were completely healed by end of treatment
 - (Day 49) 100 % of all hamsters were completely healed and remained healthy
 - (Day 49) 0 % of all hamsters had recurrence of the lesion
- CDDP-IP:
 - (Day 14) 67 % of all hamsters in this group were completely healed by end of treatment
 - (Day 49) 67 % of all hamsters were completely healed and remained healthy
 - (Day 49) 33 % of all hamsters had recurrence of the lesion

These observations are reflected graphically in Figure 4B.

0. Line 519-- “with the pretreatment biopsies (Error! Reference source not found., 6B)”, need to correct reference citation here.

Response: We thank the Reviewer for his/her catch. This issue is now corrected.

1. Please provide T stage of patients enrolled in the trial.

Response: Of patients treated in the evaluable population, 2 were T1s and 6 T2s. Two patients (#9 and #10) were excluded from the evaluable population, but included in safety analyses (safety population).

Patient	Clinical T-Stage
1	T2
2	T2
3	T1
4	T2
5	T1
6	T2
7	T2
8	T2
9*	T4
10**	T2

* Excluded from evaluable population due to tumor outside of inclusion criteria.

** Excluded from evaluable population due to concurrent documented malignancy.

This information is reported Page 9: “.....During a monitoring visit, it was determined that one subject’s tumor stage and size was outside of the eligibility criteria. Another subject had evidence of concurrent malignancy of the oral cavity and metastatic disease identified on the baseline CT examination. As such, these patients were excluded from the evaluable population, but included in safety analyses (safety population). Eight of the remaining subjects (two patients with T1 and 6 patients with T2 stage tumors) met the inclusion criteria.....”

Reviewer #3:

Editorial note – please note that Reviewer #3 (with expertise in biostatistics and clinical trial study design) did not provide any Remarks to the Author. However, while the referee highlights that the study design is appropriate, they have confidentially raised a concern on the very few sample size and the very large effect size defined (85% vs. 30% in response rate). Please explain why a smaller effect size (e.g., 60% vs. 30%) was not chosen.

Response: We thank the reviewer for this comment. Initially, the protocol was comparing a smaller effect size (65% vs. 30%). Per the protocol, once a minimum of 7 responses were achieved, the trial would be considered a success for efficacy. Seven out of the 8 evaluable patients showed a response, thus the statistical assumptions were updated in an amended protocol to reflect the observed clinical response (85% vs. 30%).

Reviewer #4 (Remarks to the Author): with expertise in oral cancer, animal models

In this paper, the authors demonstrated the efficacy and safety of a patch system to deliver cisplatin. In different animal models, they showed that the patch was therapeutically useful to treat tumors, inhibit recurrences and the development of premalignant lesions, with no toxicity. They also performed a clinical trial in HPV negative patients with positive results. Their findings will help to improve cisplatin delivery and will enhance other therapies in combination with this system. I have some comments to the authors that I would like them to address.

We thank the reviewer for his/her appreciation of our study and valuable comments that improved our paper.

* In the introduction section:

1. Clarify the first time you use IV writing intravenously (IV) as you did with other abbreviations.

Response: We thank the Reviewer for his/her notice. Corrected and recommended.

* In Materials and Methods:

2. In chemicals and reagents, DMBA and mineral oil are lacking. They are mention in the premalignant lesion animal model paragraph but I think they have to be mentioned with the other reagents.

Response: DMBA was purchased from Sigma-Aldrich and mineral oil was purchased from Millipore. The Methods section is now updated.

3. "All animal procedures were performed in accordance with a protocol approved by the Institutional Animal Care and Usage Committee": I think this statement should be located before the description of all the procedures made in the animals.

Response: Revised as suggested by the reviewer.

Also, it is important to mention the number and date of approval of the animal protocols presented and the animal welfare guidelines used to perform these experiments.

Response: All studies were conducted under protocol 1112-115-15 (approved in 2014) and the welfare guidelines were per MIT's Division of Comparative Medicine. We have updated the text as requested by the Reviewer. Page 4: "*All animal procedures were performed in accordance with National Institutes of Health Animal Care guidelines and Massachusetts Institute of Technology, Division of Comparative Medicine requirements, under a protocol 1112-115-15 (2014) approved by the Institutional Animal Care and Usage Committee.*"

Particularly, I was wondering the final humane endpoint used related to tumor volume. That is important related to the animal's welfare, as the authors worked with "human-sized tumors".

Response: The intent behind the language "Human-sized" tumors was to emphasize that the hamster cheek pouch model can grow tumors much larger than, for example, a murine model and tumors can approach 1 cm in diameter, similar to small T1 tumors in humans.

4. How biopsies of premalignant lesions were performed to confirm dysplasia?

Response: A subset of animals treated with DMBA for 4 weeks were sacrificed, and biopsies of the cheek pouch tissues were removed for histological examination to confirm generation of dysplastic lesions. The remaining animals were evaluated by palpation of the pouch surface and visually assessed for leukoplakia at 2X magnification before randomization into 3 treatment groups (as described in the Material and Methods). The revised version of the manuscript now includes the following (Page 5): "*A subset of animals were sacrificed, biopsies of the cheek pouch lesions were removed for histological examination, and generation of dysplastic lesions was confirmed by the senior veterinary pathologist. The remaining animals were randomized into 3 groups.....*."

5. A comment: Why the authors anesthetize the animals during their treat with DMBA? I would like to share with the authors that it is possible to apply DMBA topically with a syringe (without needle) with the animal

awake. The procedure is quick and the animal does not experience discomfort.

Response: We thank the Reviewer for his/her advice. The well-established protocol used in our study involves pulling out the hamster's cheek pouch prior to DMBA application. As such, animals were anesthetized to reduce stress and discomfort following the recommendation from a veterinarian.

6. In hamster, intravenous injection can be performed under anesthesia making a small incision in the skin of the neck, exposing the jugular vein. In this sense the sentence "Of note, due to the inability to perform tail vein CDDP-IV administration in hamsters, CDDP-IP delivery was used instead of the IV injection" should be removed.

Response: We thank the Reviewer for the useful advice. As treatment was administered twice a week, the suggested route of administration would require securing a jugular vein catheter system onto the neck of the hamster, in order to prevent frequent incisions. Placing a jugular vein catheter system at the dorsal nape of the neck requires expertise to be performed properly, and may endure additional stress and risk to the animals. As such, following discussion with animal care professionals and head and neck cancer expert consortium a frequently used IP delivery method was selected for systemic drug administration.

Authors believe that providing a brief explanation on why IP delivery was used in the hamster model is important for clarity and transparency. Given the Reviewer's suggestion, the above sentence was revised and in now reads as following (Page 8): "Of note, given that hamsters do not have a readily accessible tail vein, and jugular vein administration is invasive and may endure additional stress to the animals, CDDP-IP delivery was used instead of the IV injection."

7. Please, unify how many applications were performed (5 or 6?), in the following paragraph: "Hamsters were treated for 17 days (6 treatments total). Animals treated with PRV111 demonstrated substantial reduction in tumor sizes after just 3 treatments (day 10) (Figure 2B), with complete tumor regression achieved in most animals by day 17 (after 5 treatments with PRV111)".

Response: Six treatments were performed overall. A substantial reduction in tumor volume was reached after 3 treatments (at day 10), while complete tumor regression was achieved in most animals by day 17 (after 5 treatments with PRV111)". Please see Figure 2B below with treatment days indicated.

8. In the hamster model with precancerous lesions, when it says "Biopsies of lesions were collected from untreated animals for histological confirmation of dysplasia" corresponds to the control group, cancerized with no treatment?

Response: The reviewer is correct. We have revised this part to improve clarity. Page 5: "To induce precancerous lesions, hamsters were anesthetized and the left cheek pouch was painted with 1% w/v DMBA solution (Sigma-Aldrich) in mineral oil (Millipore) three times per week for 4 weeks. To limit local diffusion of the carcinogen, a small diameter sable brush was used for DMBA application. Biopsies of the cheek pouch lesions were removed for histological examination, and generation of dysplastic lesions was confirmed by the senior veterinary pathologist."

Please correct "CO2". **Response:** Corrected.

How the authors assessed toxicity after treatment? which parameters they evaluated? for example, mucositis development?

Response: (i) Overall health of the animals (food and water intake, abnormal stool/urine, moist fur or mouth, rapid or labored breathing, physical appearance and behavior) were monitored daily throughout the treatment period, as described in the manuscript (Page 4). (ii) Body weights were monitored as shown on Figure 4C. (iii) At the end-point, major organs were isolated, and tissue distribution of cisplatin was evaluated, showing only negligible level of the drug in body organs following PRV111 application (Figure 4E). (iv) Evaluation of kidney tubular necrosis was performed upon necropsy, showing no damage to kidney in animals treated with PRV111. Histological images are provided on Figure 4F. (v) Local toxicity (gross examination and histopathology) was also evaluated. No signs of local irritation, redness, or inflammation were observed at the PRV111 application site, as reported in the manuscript. Page 9: "*There was no inflammation or other damage detected at the site of the PRV111 application (Supplementary Figure 5B), and no local toxicity was observed in any other area of the oral cavity, including the tongue and or other cheek pouch.*"

9. *In vivo* permeation study was performed in porcine tissue, the animals should be mentioned in the text and a description should be added.

Response: Revised as suggested by the Reviewer. Page 5: "*For experiments using normal porcine buccal tissue, specimens were collected immediately after animal was sacrificed. The remaining steps were identical to those used to treat oral tumors in a living hamster. Due to its close resemblance to human buccal mucosa in structure, enzyme activity and permeability characteristics, porcine buccal mucosa has been extensively used as a model to study the permeability of various diffusants, and to assess their potential to be delivered through the buccal route³¹. Since human oral mucosa is not widely available, normal porcine buccal mucosa was used to assess permeation of nanoparticles loaded to PRV111.*" Legends for Supplementary Figure 4: "*As FDA requires a large animal model to evaluate drug delivery devices prior to initiating human clinical studies, porcine oral mucosa was used for this experiment. Representative fluorescence images of the freshly harvested normal porcine buccal tissue sections taken after treatment of the specimen with PRV111 patch containing chitosan particles labeled with Cy5 (red) and encapsulating FITC (green). Right – PE was applied prior to PRV111; left – PE was not applied.*"

Comment #1: Why did the authors study *in vivo* permeation in porcine tissue and not in the hamster normal pouch?

Response: The FDA requires a large animal model to evaluate drug delivery devices prior to initiating human clinical studies. Porcine oral mucosa is non-keratinized and resembles that of humans more closely than any other non-primate animal in terms of structure and composition. Since human oral mucosa is not widely available, porcine buccal mucosa is frequently used as model for transmucosal studies (PMID: 18506782, PMID: 22309108). As requested by the Reviewer, we have now added additional text and references to the revised version of the manuscript. Please see our response to the previous comment.

Comment #2: Why the authors did not assess the permeation treatment in oral precancerous lesions?

Response: While surface dysplastic lesions can be dissected from subepithelial tissues for pathological evaluation, these lesions are thin, and can't be used to permeation assessment.

* In the Results section:

0. There are no control groups, ie, animals with tumors without any treatment. Those models are well reported in previous studies?

Response: Preclinical tumor models used in our work are well established and have been used to study head and neck tumorigenesis for a few decades. Nevertheless, we have included multiple control groups for every *in vivo* experiment performed in this study. Specifically, for FaDu subcutaneous tumor model two control groups were incorporated: (i) IT injection of PBS, and (ii) IT administration of blank NPs without cisplatin encapsulation. For the orthotopic hamster model we also used 2 control groups: (i) IP injection of PBS, and (ii) drug-free blank patch. As such, all experiments presented in this manuscript are fully controlled.

For the case of the oral precancerous lesion model, the cited bibliography does not match exactly with the

cancerization protocol, have you characterized this model previously?

Response: The protocol used in this study to induce oral precancerous lesions that are histologically similar to the lesions in humans was established in 1970s (PMID: 4557243), and slightly different adaptations of this well characterized method are frequently used to study oral tumorigenesis to this date (PMID: 26892902, PMID: 20157056, PMID: 21484926, PMID: 23200011, PMID: 30620118, PMID: 24179524, PMID: 22848241, etc). We have updated the references to include additional studies describing this approach.

2. In the text, it says, “Animals in the CDDP-IV group were euthanized at day 7 after treatment initiation due to the rapidly decreasing body weight and signs of toxicity associated with high level of systemic CDDP circulation (Figure 1B)”. It has to say, “figure 1E”.

Response: We thank the Reviewer for this catch. Corrected.

3. In Supplementary Figure 1 legend, it says: “...Immunodeficient nude mice bearing subcutaneous human HNSCC FaDu cell line induced tumors treated with IP administration of CDDP-NPs”. If I understood correctly, no IP injection was performed in this model.

Response: We thank the Reviewer for his/her note. Corrected.

4. In figure 2E, the authors should explain in the figure legend what “yellow” fluorescent in the photo means.

Response: Addressed as suggested by the Reviewer. Page 17: “*Yellow areas display dual-labeling, NPs with encapsulated FITC.*”

5. In the paragraph “PRV111 increased the number of tumor-infiltrating lymphocytes” line 519, correct the error.

Response: We thank the Reviewer for his/her catch. This issue is now corrected.

6. Could the authors clarify the following paragraph?: “Topical administration of PRV111 reduced cisplatin associated toxicities”, it says that “Analysis performed one hour after administration of the first dose demonstrated that blood level of cisplatin in the CDDP-IP group was over 20-fold higher than the level detected in PRV111 treated animals (Figure 3A). Hamsters were sacrificed and major organs such as liver, lungs, kidneys, spleen, brain and heart, as well as the tongue, healthy contralateral cheek pouch, and tumors, were isolated to evaluate the tissue distribution of cisplatin in PRV111 and CDDP-IP treated groups using ICP-MS. Note, we were able to collect very small residual tumors samples from three animals treated with PRV111, as the remaining hamsters were tumor free at the end point”. Which endpoint the authors considered?

Response: We have slightly rephrased this paragraph for clarity. Page 8: “*Analysis performed one hour after administration of the first dose demonstrated that blood level of cisplatin in the CDDP-IP group was over 20-fold higher than the level detected in PRV111 treated animals (Figure 3A). At the end-point, hamsters were sacrificed and major organs such as liver, lungs, kidneys, spleen, brain and heart, as well as the tongue, healthy contralateral cheek pouch, and tumors, were isolated to evaluate the tissue distribution of cisplatin in PRV111 and CDDP-IP treated groups using ICP-MS. Note, we were able to collect very small residual tumors samples from three animals treated with PRV111, as the remaining hamsters were tumor free upon necropsy.*”

It is not clear which tumors treated with PRV111 were evaluated as in the long-term recurrence study none of the animals developed recurrences so no tumors were available.

Response: For the long-term recurrence study, a different group of animals bearing HCPC-1 induced orthotopic tumors was treated with a full course of PRV111 therapy or IP cisplatin injections. Six animals that achieved complete tumor regression after treatment with PRV111 and six animals that achieved complete response after treatment with CDDP-IP were used for the long-term recurrence study (3 animals from each group were used for planned euthanasia experiment, and 3 animals from each group were monitored until death).

7. Figure 5 supplementary: Is it possible for the authors to provide a better photo representing a precancerous lesion in the hamster cheek pouch?

Response: As requested by the Reviewer, we have added additional image to Supplementary Figure 5.

* In the Discussion section:

17. Comment: The fact that PRV111 was effective in treating HPV negative tumors is important, as it is well known that HPV positive tumors respond better to radiotherapy than HPV negative tumors. In this sense, do the authors think possible to combine this strategy with radiation therapy to enhance HPV negative tumor response.

Response: Certainly, the potential clinical utility of PRV111 to deliver cisplatin locally has a number of potential applications. Systemic cisplatin when administered concurrent with radiotherapy is associated with a survival benefit driven primarily by radiosensitizing effects reflected in reduction in locoregional recurrence (pingon meta-analysis). As such, there is potential interest in combining PRV111 for topical cisplatin delivery in combination with radiation therapy to optimize radiosensitization locally while reducing systemic toxicity related to current systemic cisplatin paradigms. Given the relative radioresistance of HPV negative HNSCC as compared with HPV positive disease, there is substantial interest in developing PRV111 in the context of improving outcomes for HPV negative disease in combination with radiation therapy.

REVIEWERS' COMMENTS

Reviewer #1 (Remarks to the Author):

The authors have addressed all issues. Thus, I enthusiastically support the acceptance of this work.

Reviewer #2 (Remarks to the Author):

I would like to thank the authors for the detailed response.

Pertaining pathologic response / percentage of viable tumor cells, the suggestion is not that this becomes the study primary endpoint, but at least an exploratory endpoint. It would be important to report the pathologic findings/tumor bed histopathologic features; for instance, if there were no viable tumor cells in the surgical specimen that would be very valuable to know as this is highly prognostic.

Reviewer #4 (Remarks to the Author):

Thank you very much to the authors that have addressed in detail all the suggestions made. Their explanations were very clear.

I have a couple of comments.

-Is it possible to adapt and include this response to the reviewer in M&M? I think it could be interesting for someone who is not familiar with the hamster model.

"The intent behind the language "Human-sized" tumors was to emphasize that the hamster cheek pouch model can grow tumors much larger than, for example, a murine model and tumors can approach 1 cm in diameter, similar to small T1 tumors in humans"

-Thank you for your reformulation related to the statement: Of note, given that hamsters do not have a readily accessible tail vein, and jugular vein administration is invasive and may endure additional stress to the animals, CDDP-IP delivery was used instead of the IV injection. Could you please move this paragraph to the M&M? I think it is better to be there than in the results section.

-Related to the in vivo permeation study that was performed in porcine tissue. Thank you very much for your clarification. As it is important that you included a study in large animals I think it is worth including information related to the procedures made in this animal model (in terms of anesthesia, patch application technique, etc). Do you have a particular ethics protocol to mention related to pigs? It would be interesting to mention why you did not assess permeation in the cancerized pouch. After this paragraph: "Note, nanoparticles loaded to PRV111 were designed to penetrate and absorb only in areas where PE (a critical component of the PRV111 system) is applied, thus minimizing the damage to the adjacent non-tumorous tissue (Supplementary Figure 4)". I think is important to mention, after this paragraph, that this experiment was performed in pigs and why, as it is in the hamster section.

Related to your response to: "For the case of the oral precancerous lesion model, the cited bibliography does not match exactly with the cancerization protocol, have you characterized this model previously?"

It is true that there are different adaptations of this cancerization method. However, number of applications / week, total number of applications, vehicle in which you dissolved DMBA could modify

the toxicity of the protocol and the aggressiveness of the oral precancer model in terms e.g. when the oral lesions appear, the number of animals that exhibit oral precancerous lesions, type of lesions, the development of tumors etc.

My suggestion was to cite a publication that could be similar to your protocol, i.e. topical application of DMBA in mineral oil, 3 times a week during 4 weeks or at least a protocol that equals the number of applications (12 applications).

REVIEWERS' COMMENTS

Reviewer #1 (Remarks to the Author):

The authors have addressed all issues. Thus, I enthusiastically support the acceptance of this work.

Response: We thank the Reviewer for his/her comments that helped improve our study.

Reviewer #2 (Remarks to the Author):

I would like to thank the authors for the detailed response. Pertaining pathologic response / percentage of viable tumor cells, the suggestion is not that this becomes the study primary endpoint, but at least an exploratory endpoint. It would be important to report the pathologic findings/tumor bed histopathologic features; for instance, if there were no viable tumor cells in the surgical specimen that would be very valuable to know as this is highly prognostic.

Response: We thank Reviewer for his/her thoughtful input. As indicated in the manuscript, after 4 treatment visits with PRV111 (all in a week), a significant clinical response was achieved in 7 of 8 subjects in evaluable population, with 6 PR and 1 CR (Supplementary Table 6). As reviewer #2 astutely noted, the study protocol did include a "Histopathologic Response to Treatment" as a secondary endpoint. The histologic assessment of the tumor bed region demonstrated significant tumor necrosis caused by the PRV111 treatment. **Supplementary Figure 6** in the revised version of the manuscript shows representative H&E stained histologic images of tumor bed areas post-PRV111 treatment, demonstrating marked ulceration, necrosis and granulation tissue formation at the site of patch placement.

Reviewer #4 (Remarks to the Author):

Thank you very much to the authors that have addressed in detail all the suggestions made. Their explanations were very clear. I have a couple of comments.

* Is it possible to adapt and include this response to the reviewer in M&M? I think it could be interesting for someone who is not familiar with the hamster model. "The intent behind the language "Human-sized" tumors was to emphasize that the hamster cheek pouch model can grow tumors much larger than, for example, a murine model and tumors can approach 1 cm in diameter, similar to small T1 tumors in humans".

Response: Done as requested. Page 4: "*Note, since in this model PRV111 was applied topically to the human-sized tumors (hamster cheek pouch tumors can approach 1 cm in diameter, similar to small T1 tumors in humans), cisplatin dosage was based on tumor surface area (cm²), rather than the body surface area (BSA) or a body weight.*"

* Thank you for your reformulation related to the statement: "Of note, given that hamsters do not have a readily accessible tail vein, and jugular vein administration is invasive and may endure additional stress to the animals, CDDP-IP delivery was used instead of the IV injection". Could you please move this paragraph to the M&M? I think it is better to be there than in the results section.

Response: Done as requested.

* Related to the in vivo permeation study that was performed in porcine tissue. Thank you very much for your clarification. As it is important that you included a study in large animals I think it is worth including information related to the procedures made in this animal model (in terms of anesthesia, patch application technique, etc). Do you have a particular ethics protocol to mention related to pigs?

Response: The information requested by the reviewer is already included on the Page 6 of the manuscript: "*For experiments using normal porcine buccal tissue, specimens were collected immediately after animal was sacrificed. The remaining steps were identical to those used to treat oral tumors in a living hamster*".

* It would be interesting to mention why you did not assess permeation in the cancerized pouch.

Response: It is not entirely clear what Reviewer means by "cancerized pouch", but we assume he/she is referring to the precancerous lesions. As we have described in our previous Response letter, surface dysplastic lesions are thin, and can't be used to permeation assessment. As our results clearly demonstrate deep permeation and wide distribution of the NPs throughout the thick overt tumor mass, it is obvious that they permeate through a very superficial dysplastic lesion.

* After this paragraph: "Note, nanoparticles loaded to PRV111 were designed to penetrate and absorb only in areas where PE (a critical component of the PRV111 system) is applied, thus minimizing the damage to the

adjacent non-tumorous tissue (Supplementary Figure 4)". I think is important to mention, after this paragraph, that this experiment was performed in pigs and why, as it is in the hamster section.

Response: Done as requested. Page 8: *"Note, nanoparticles loaded to PRV111 were designed to penetrate and absorb only in areas where PE (a critical component of the PRV111 system) is applied, thus minimizing the damage to the adjacent non-tumorous tissue (Supplementary Figure 4). As FDA requires a large animal model to evaluate drug delivery devices prior to initiating human clinical studies, porcine oral mucosa was used for this experiment."*

* Related to your response to: "For the case of the oral precancerous lesion model, the cited bibliography does not match exactly with the cancerization protocol, have you characterized this model previously?". It is true that there are different adaptations of this cancerization method. However, number of applications / week, total number of applications, vehicle in which you dissolved DMBA could modify the toxicity of the protocol and the aggressiveness of the oral precancer model in terms e.g. when the oral lesions appear, the number of animals that exhibit oral precancerous lesions, type of lesions, the development of tumors etc. My suggestion was to cite a publication that could be similar to your protocol, i.e. topical application of DMBA in mineral oil, 3 times a week during 4 weeks or at least a protocol that equals the number of applications (12 applications).

Response: We thank the Reviewer for the explanation. The cited literature indeed extensively covers this well-characterized and frequently used model.